# Erratic and blood vessel-guided migration of astrocyte progenitors in the cerebral cortex

Hidenori Tabata [1,2] ✉, Megumi Sasaki[2], Masakazu Agetsuma [3], Hitomi Sano [2], Yuki Hirota [2], Michio Miyajima[2], Kanehiro Hayashi[2], Takao Honda[2], Masashi Nishikawa[1], Yutaka Inaguma[1], Hidenori Ito[1], Hirohide Takebayashi [4], Masatsugu Ema[5], Kazuhiro Ikenaka[6], Junichi Nabekura [3], Koh-ichi Nagata [1] & Kazunori Nakajima [2] ✉

Astrocytes are one of the most abundant cell types in the mammalian brain. They play essential roles in synapse formation, maturation, and elimination. However, how astrocytes migrate into the gray matter to accomplish these processes is poorly understood. Here, we show that, by combinational analyses of in vitro and in vivo time-lapse observations and lineage traces, astrocyte progenitors move rapidly and irregularly within the developing cortex, which we call erratic migration. Astrocyte progenitors also adopt blood vessel-guided migration. These highly motile progenitors are generated in the restricted prenatal stages and differentiate into protoplasmic astrocytes in the gray matter, whereas postnatally generated progenitors do not move extensively and differentiate into fibrous astrocytes in the white matter. We found Cxcr4/7, and integrin β1 regulate the blood vessel-guided migration, and their functional blocking disrupts their positioning. This study provides insight into astrocyte development and may contribute to understanding the pathogenesis caused by their defects.

Astrocytes are major components of the mammalian brain. They play essential roles in maintaining and regulating synapses, blood brain barrier, blood flow, and metabolism in the mature brain[1–4]. During development, astrocytes are generated from radial glia, mainly after neurogenesis has ended. They migrate into the cortical plate (CP) at the time of neural network formation and actively participate in synapse formation, maturation, and elimination[5–7]. Therefore, astrocyte positioning at the right place and time is important for proper neural network formation. However, how astrocytes migrate and settle in the CP is largely unknown because of the lack of specific markers for astrocyte progenitors and the numerous cell divisions they undergo

after leaving the ventricular zone (VZ). The most accepted model of astrocyte migration proposes that, after the neurogenesis stage, radial glial cells retract their ascending processes, which reach the pial surface, to get the cell body out from the VZ. This model was established based on observations that radial glia, labeled with fluorescent dyes injected under the pial surface of newborn ferrets, transformed and ultimately differentiated into glial fibrillic acidic protein (GFAP)-positive astrocytes in the white matter[8]. However, since GFAP is virtually only expressed in the white matter and layer I under the pia mater, whether protoplasmic astrocytes from the cortical gray matter, which are the main regulators of synapse formation, migrate in the

[1]Department of Molecular Neurobiology, Institute for Developmental Research, Aichi Developmental Disability Center, 713-8 Kamiya, Kasugai, Aichi 480-0392, Japan. [2]Department of Anatomy, Keio University School of Medicine, 35 Shinanomachi, Shinjuku-ku, Tokyo 160-8582, Japan. [3]Division of Homeostatic Development, National Institute for Physiological Sciences, 38 Nishigohnaka Myodaiji-cho, Okazaki, Aichi 444-8585, Japan. [4]Division of Neurobiology and Anatomy, Graduate School of Medical and Dental Sciences, Niigata University, 1-757 Asahimachi, Chuo-ku, Niigata 951-8510, Japan. [5]Department of Stem Cells and Human Disease Models, Research Center for Animal Life Science, Shiga University of Medical Science, Seta, Tsukinowa-cho, Otsu, Shiga 520-2192, Japan. [6]Division of Neurobiology and Bioinformatics, National Institute for Physiological Sciences, 5-1 Higashiyama, Myodaiji, Okazaki 444-8787, Japan. ✉e-mail: tabata@inst-hsc.jp; kazunori@keio.jp

same way remains unknown. Recent studies indicated the existence of subventricular zone-derived astrocytes[9,10], however, the cellular and molecular mechanisms for their migration are still obscure.

Here, we labeled cells derived from the cortical VZ in the late stages of CP development in mice by in utero electroporation of fluorescent protein-expression vectors, and identified a characteristic migration mode, named erratic migration, in which cells move rapidly and almost randomly within the intermediate zone and the CP. Cell fate analysis of erratically migrating cells reveal that they are astrocyte progenitors. We also find astrocyte progenitors migrate along blood vessels and spread in the CP using both erratic and blood vessel-guided migrations. These blood vessel-independent (erratic) and vessel-guided migrations can be observed in living mouse embryos by two-photon in vivo imaging. These two migration modes are seen only for astrocyte progenitors derived from prenatal VZ, which eventually differentiate into protoplasmic astrocytes within the cortical gray matter, whereas progenitors derived from postnatal VZ remain in the white matter and differentiate into fibrous astrocytes. We further find that astrocyte progenitors that bridge neighboring blood vessels may support angiogenesis itself. Moreover, we uncover the chemokine receptors, Cxcr4 and Cxcr7, and the downstream integrin β1 are involved in blood vessel-guided migration. Indeed, their functional blocking reduce the astrocyte density in the superficial cortical gray matter, indicating the importance of blood vessel-guided migration for proper positioning of astrocytes. Altogether these data provide fundamental mechanisms of astrocyte migration and positioning in the developing cerebral cortex.

## Results

### Erratic migration: a characteristic migration mode of astrocyte progenitors

Astrocytes and neurons are directly or indirectly generated from progenitors in the VZ. We investigated the migratory behavior of all types of cells arising from the cortical VZ by time-lapse imaging of brain slices[11,12]. VZ cells were transfected at embryonic day (E) 15 with a CAG promoter (non-selective promoter) driven-green fluorescent protein (GFP) expression vector (CAG-EGFP) by in utero electroporation[13], and cortical slices were employed to time-lapse imaging at E17. We found that some GFP-positive cells migrated rapidly and almost randomly within the intermediate zone (IZ) and CP (Fig. 1a, b; Supplementary Movie 1). These cells divided during the migration, and the daughter cells exhibited the same irregular pattern (Fig. 1a, 15.3 h). Since this migration profile was different from that of the well-known radial migration of neurons (Fig. 1a, b; Supplementary Movie 1), we named it erratic migration. Cells undergoing erratic migration (hereafter referred to as "erratically migrating cells") moved more quickly than those undergoing radial migration (Fig. 1c). Erratically migrating cells frequently changed direction (Supplementary Fig. 1a), although they ultimately tended to move toward the brain surface (Supplementary Fig. 1b). To determine the fate of erratically migrating cells, we labeled them with a photo-convertible protein, KikGR, whose fluorescence changes from green to red in response to irradiation with a 405-nm laser[14]. We electroporated with pCAG-KikGR at E15 or E16, and performed time-lapse observation 24 h later. After identifying the cells undergoing either erratic or radial migration in slices, we irradiated them with a 405-nm laser (Fig. 1d, Supplementary Fig. 1c), dissociated the slices, and cultured the cells on coverslips for 4 days. As a result, most of the erratically migrating cells differentiated into GFAP-positive astrocytes (15 out of 17 clones/ 17 slices from 13 brains from 3 independent cultures), whereas the cells that had undergone radial migration became β tubulin III (TuJ1)-positive neurons (7 out of 7 clones/ 5 slices from 5 brains from 2 independent cultures) (Fig. 1e). These results indicated that erratically migrating cells differentiated into astrocytes in vitro. Considering that differentiation might be affected by conditions in dissociated culture, we then asked whether astrocyte progenitors or neurons used erratic migration in organotypic slice culture. To this end, we labeled each cell type with specific

promoters (Fig. 1f). Neurons were labeled with an α-tubulin (Tα1) promoter-driven GFP expressing vector (Tα1-EGFP). To label astrocyte progenitors, we used a Cre-dependent red fluorescent protein (RFP)-expressing PiggyBac transposon vector (pPB-CAG-LNL-turboRFP)[15,16], which is integrated into the host genome with the help of a transposase-expressing vector (pCAG-hyPBase), and can efficiently label proliferative astrocyte lineage cells[9,17,18]. Cre was expressed under the human GFAP promoter (hGFAP-Cre), which is preferentially active in the astrocyte lineage. Since the hGFAP promoter is also active in a sub-population of VZ cells[19,20], cells with hGFAP promoter-active (RFP⁺) and Tα1 promoter-inactive (GFP⁻) were considered astrocyte progenitors. We mixed and electroporated these plasmids at E15, and carried out time-lapse observations 3 days later (Fig. 1f). We observed that many RFP⁺/GFP⁻ cells migrated in the erratic migration mode especially in the IZ, while GFP⁺ neurons migrated radially as previously established (Fig. 1g, Supplementary Movie 2). Within the CP, the angles between the moving directions of RFP⁺/GFP⁻ cells and the radial direction toward the brain surface were significantly wider than those of GFP⁺ cells (Fig. 1h). These observations confirmed that erratic migration was the characteristic migration mode of astrocyte progenitors.

### Cortical VZ-derived Olig2-positive cells differentiate into protoplasmic astrocytes in the cortical gray matter in vivo

Next, we mapped the in vivo fate of the erratically migrating cells. We previously reported the existence of slowly (SEP) and rapidly (REP) exiting populations of cortical VZ-derived cells, and that REP included Olig2-positive cells[21]. Olig2 is a basic helix–loop–helix transcription factor, and is known to be expressed in astrocyte progenitors in the developing cortex[9,10,22], suggesting that Olig2 might be expressed in erratically migrating cells. In fact, when we electroporated with CAG-EGFP at E15.5 and fixed at E17, GFP/Olig2-double-positive cells were found to extend short leading processes in various directions (Supplementary Fig. 1d), presenting a similar morphology to that of the erratically migrating cells we observed in time-lapse observations (Fig. 1a). To confirm their Olig2 expression more directly, we performed time-lapse observations followed by immunohistochemical analyses of the erratically migrating cells for Olig2. E15.5 embryos were electroporated with Cre dependent expression vectors pCAG-LNL-EGFP and pCAG-LNL-LynEGFP, as well as Nestin promoter driven Cre expression vector (Nestin-Cre), and prepared slices 2 days later. Nestin-Cre was used to label astrocyte progenitors preferentially. LynEGFP is a membrane associated EGFP fused with partial amino acid sequence of Lyn to visualize entire cell morphology[23]. We found that 83% of erratically migrating cells (28 out of 34 cells, in 10 brain slices) were Olig2 positive [Fig. 2a, b; this percentage may be underestimated, because of the problem of accessibility of Olig2-antibody for thick (250 μm) slices], indicating that at least a major population of erratically migrating cells were Olig2 positive. Taking advantage of this fact, to analyze in vivo fate of erratically migrating cells, we used Olig2-CreER; Z/EG double heterozygous mice, in which Cre recombinase is activated in Olig2-expressing cells upon tamoxifen administration[24], and GFP is expressed in Cre-activated cells[25]. To exclude Olig2-positive oligodendrocyte progenitor cells (OPCs) or GABAergic neurons derived from the medial ganglionic eminence, we electroporated pCAG-hyPBase and pPB-CAG-LNL-turboRFP into cortical VZ cells of these mouse embryos at E15. We administered tamoxifen at E17, and examined GFP/turboRFP double-positive cells, which were assumed to be Olig2-lineage cells derived from the cortical VZ (as at E17) (Fig. 2c). The analysis of the brains at P30 (Fig. 2d) revealed that 80.1% of double-positive cells were Aldh1l1-positive astrocytes and 19.9% were glutathione S-transferase π (GSTπ)-positive oligodendrocytes (Fig. 2e–g, 151 cells in 18 slices from 3 brains). Interestingly, all GFP/turboRFP/Aldh1l1-positive cells were found in the cortical gray matter and assumed the typical morphology of protoplasmic astrocytes extending highly branched bushy processes around the cell body

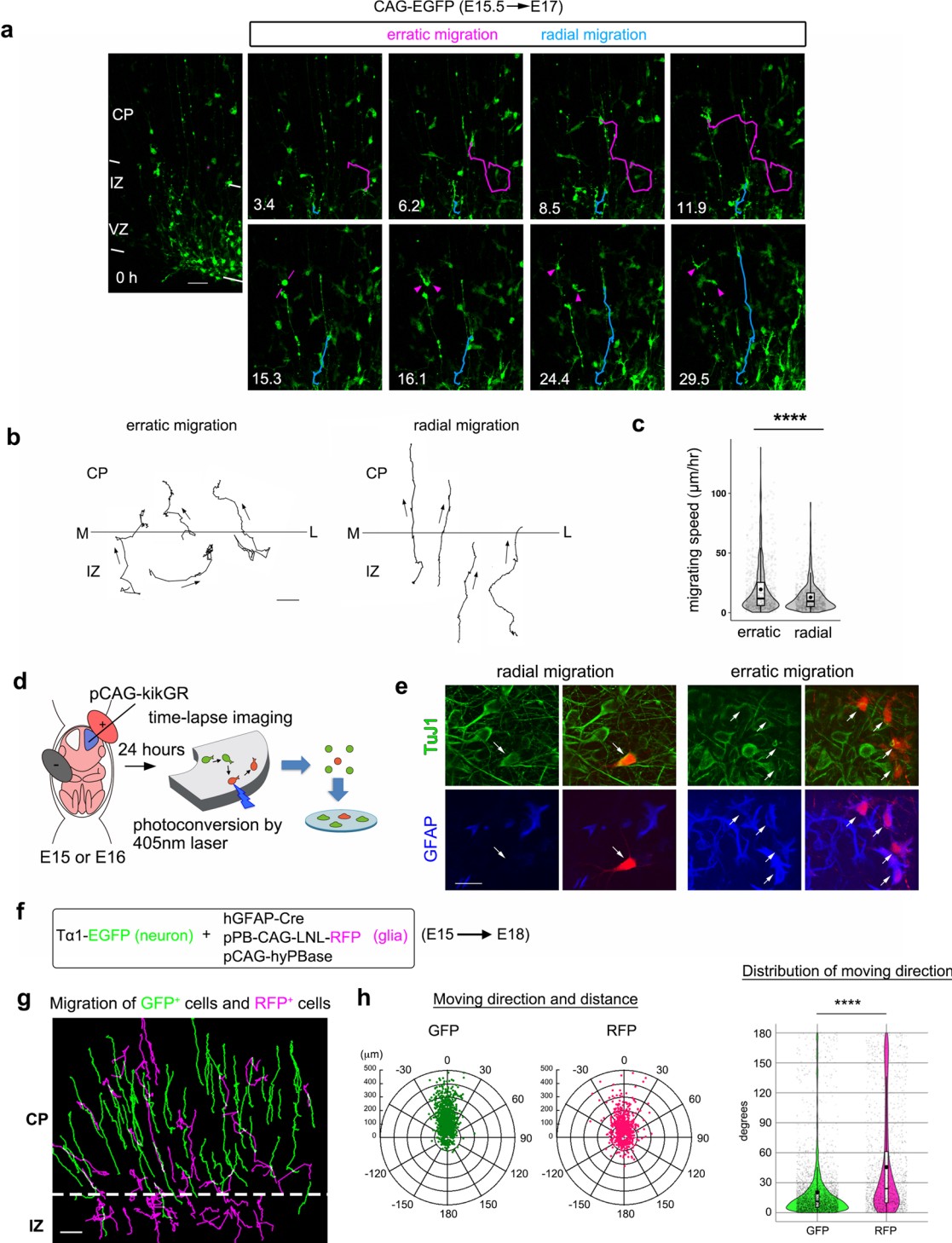

(Fig. 2e, h), indicating that the major population of the cortical VZ-derived Olig2+ cells differentiated into protoplasmic astrocytes in the cortical gray matter. As alternative in vivo fate mapping methods, we used the targeted integration with linearized dsDNA-CRISPR (Tild-CRISPR) method[26]. This method enabled us to integrate the Cre expression cassette into the *Olig2* gene in cortical VZ cells in utero. When the pPB-CAG-LNL-EGFP vector was simultaneously introduced, GFP was mostly detected in S100 calcium-binding protein B (S100β)- and/or GFAP-positive astrocytes (Supplementary Fig. 2a–d, 94% from 50 GFP-positive cells), confirming that the major population of the cortical VZ-derived Olig2+ cells differentiated into astrocytes including protoplasmic astrocytes in the cortical gray matter.

Astrocytes are generated not only in late embryonic but also in early postnatal stages[10]. A previous study demonstrated that electroporation of a conventional GFP expression vector (pCAG-EGFP) at P0–2 labels mostly astrocytes in the white matter and very few in the cortical gray matter[27]. However, because of their active cell division, astrocytes are not well labeled with this conventional plasmid vector[28], and further investigations of the final localization of astrocytes derived from the cortical VZ at respective stages remain to be carried out. To address this issue, we systematically electroporated pPB-CAG-EGFP at E15, E17, or P1, and fixed the brains at P8. We introduced pCAG-turboRFP simultaneously as a reference. When we performed electroporation at E15, layer II/III neurons and protoplasmic astrocytes

**Fig. 1 | Erratic migration: a characteristic migration mode of astrocyte progenitors. a** Observations of erratic migration (magenta) and radial migration (blue). VZ derived cells were labeled by in utero electroporation as indicated. The slice at the beginning of observation is shown in left. The erratically migrating cell divided at 15.3 h (h) and continued erratic migration (arrowheads). See also Supplementary Movie 1. **b** Trajectories of erratic (left) and radial migration (right). The arrows indicate the direction of movement. The horizontal bars represent the border between the CP and the IZ (M, medial; L, lateral). **c** The migration speeds during each frame interval. Erratic migration is faster than radial migration (Welch t-test, 932 and 961frames of erratic (11 cells) and radial migration (10 cells) in 4 slices, $P < 0.0001$). **d** Strategy to investigate the final fate of erratically migrating cells in vitro. **e** Marker staining of radial and erratic migration cells (arrows) after the cultivation. Radial and erratically migration cells differentiated into TuJ1+ neurons (7 among 7 clones), and GFAP+ astrocytes (15 among 17 clones). Samples were pooled from three independent experiments. **f** Method for differentially labeling of

neurons (EGFP, green) and astrocyte progenitors (turboRFP, magenta). **g** Trajectories of GFP- and turboRFP-expressing cells during 25 h of observation. Erratic migration could be found only for the turboRFP-labeled population. See also Supplementary Movie 2. **h** Distances and directions related to the radial axis (0 degree is the direction from the VZ to CP) achieved by GFP- and turboRFP-expressing cells within the observation period (24.5 h). The distances and directions are plotted in polar graphs (left), and the distribution of moved directions are shown in violin and box plots (right). The moving directions of turboRFP-expressing cells were significantly broader compared to the GFP-expressing cells (Two-tailed Mann-Whitney test, 868 turboRFP+ cells vs. 1944 GFP+cells, $P < 0.0001$). CP cortical plate, IZ intermediate zone, VZ ventricular zone; Scale bars, 100 μm (**g**), 50 μm (**a**, **b**), 20 μm (**e**). For detailed information of box plots, see "Statistical analysis" section in Methods. Source data are provided as a Source Data file.

---

throughout the gray matter were labeled with RFP and GFP, respectively (Supplementary Fig. 3a-left). More GFP-positive astrocytes were found in the superficial layers of the gray matter than in deep layers (Supplementary Fig. 3c). Many GFP-positive astrocytes were localized between RFP-expressing neurons in layers II/III and presented a morphology of protoplasmic astrocytes (Supplementary Fig. 3b). After electroporation at E17, neurons were no longer labeled with RFP (Supplementary Fig. 3a-middle). GFP-labeled astrocytes were still found in the gray matter, especially in layer I, where astrocytes strongly expressed GFAP and formed glia limitans beneath the pia mater (Supplementary Fig. 3d). Within the gray matter under layer I, the density of GFP-labeled astrocytes in the layer II/III (bin 2) significantly reduced (Supplementary Fig. 3a-middle, 3c; E15 *vs.* E17, $P = 0.0045$). After electroporation at P1, GFP-positive cells were mainly observed in the white matter or in deep gray matter (Supplementary Fig. 3a-right, 3e) and most of them were strongly GFAP-positive fibrous astrocytes (Supplementary Fig. 3e), suggesting that this type of astrocytes was mainly generated postnatally. These results were consistent with those of the Tild-CRISPR experiment (Supplementary Fig. 2c, d), in which GFP-positive astrocytes were observed not only in the gray matter but also in the white matter due to the continued expression of Cre in the postnatal stages. Time-lapse experiments revealed that these postnatally labeled cells did not move extensively, unlike the prenatally labeled erratically migrating cells (Supplementary Fig. 3f, g; Supplementary Movie 3). These observations suggested that highly motile astrocyte progenitors destined to the cortical gray matter were produced mainly at prenatal stages, whereas fibrous astrocytes in the white matter were generated postnatally.

## Blood vessel-guided migration: another migration mode of astrocyte progenitors

While analyzing the trajectory of astrocyte progenitors (RFP+/GFP− cells), we noticed that a substantial number of them migrated radially, especially in the CP (Fig. 1g). We hypothesized that they migrated along blood vessels because blood vessels tend to run radially in the CP during the embryonic and perinatal stages[29] and the clonal expansion of astrocytes along blood vessels has been previously reported[30]. Hence, we investigated the relationship between blood vessels and astrocyte progenitors labeled with a plasmid mixture (pPB-CAG-LNL-GFP, hGFAP-Cre, and pCAG-hyPBase) using electroporation at E15. The brains were fixed 3 days later (E18), and we indeed observed that a subpopulation of GFP and Olig2 double-positive cells was strongly associated with blood vessels especially in the CP (Supplementary Fig. 4a). These cells contacted blood vessels with their cell bodies and/ or processes. Conversely, the blood vessel-associated cells were mostly Olig2 positive (78.1%, 274 cells from 8 brains), as were the erratically migrating cells derived from the cortical VZ (Fig. 2a, b), indicating that the lineage trace using *Olig2-CreER* mice includes these cells. As an alternative method to label astrocyte progenitors, we used

Aldh1l1-GFP transgenic mice, which express GFP in the astrocyte lineage from as early as P0. At P0, GFP-positive cells were located in the vicinity of blood vessels, especially in the superficial CP (Supplementary Fig. 4b, c). We measured the distances from GFP-positive cells as well as from all DAPI-stained cell nuclei to the nearest blood vessels in the superficial CP. There was a prominent accumulation of GFP-positive cells, and not DAPI-stained cells, near (less than 5 μm) blood vessels (Supplementary Fig. 4d). Overall, the distance from the GFP-positive cells to blood vessels was significantly smaller than that between blood vessels and DAPI-positive nuclei.

To determine whether astrocyte progenitors were migrating along or just attached to blood vessels, we performed time-lapse analyses. To label the astrocyte lineage specifically, we developed a new system, the rDIO system, by combining the DIO system[31,32], in which GFP expression depends on hGFAP promoter-driven Cre recombinase expression, with the Dre-rox system[33,34], in which the GFP cassette is excised by Dre recombinase that is expressed under the control of a neuron-specific doublecortin (Dcx) promoter (pPB-CAG-rDIO-EGFP, Fig. 3a). As expected, all GFP expressing cells observed in the cortical gray matter at P10 were astrocytes (Fig. 3b). We electroporated the rDIO system into E15 *Flt1-DsRed* mice, which express DsRed in vascular endothelial cells[35], and performed time-lapse analyzes at E17 (Fig. 3a). GFP-labeled cells were found to migrate actively along blood vessels (Fig. 3c, d, Supplementary Movies 4, 5). This blood vessel-guided migration was observed for about half of the astrocyte progenitors in the CP at this stage (Fig. 3e, 51% in 168 migrating cells). We observed that astrocyte progenitors switched vessel-independent erratic migration to blood vessel-guided migration and vice versa (Fig. 3d; Supplementary Movie 5). Interestingly, we also observed that astrocyte progenitors bridged neighboring blood vessels before the blood vessels form a new branch along the progenitors, suggesting that astrocyte progenitors support angiogenesis (Fig. 3f, Supplementary Fig. 5, Supplementary Movies 6, 7). This mutual dependency between astrocyte progenitors and blood vessels might contribute to the ubiquitous distribution of astrocytes and the formation of dense vascular networks in the developing cortex.

## Two-photon in vivo imaging of blood vessel-guided migration and blood vessel-independent migration

Since the above-mentioned migration modes of astrocyte progenitors were observed in slice cultures, we next investigated these cell behaviors in brains of intact living embryos using two-photon in vivo imaging of *Flt1-DsRed* mouse embryos whose astrocyte progenitors had been labeled with the rDIO system as in slice culture (Fig. 4a). We successfully found cells migrating on (Fig. 4b, c, Supplementary Movies 8, 9) and between blood vessels (Fig. 4d, e, Supplementary Movies 11, 12) in intact E17 brains electroporated with the rDIO system at E15. We also observed one GFP-positive cell migrated in the vicinity of a blood vessel, attached it, and then migrated on it (Supplementary

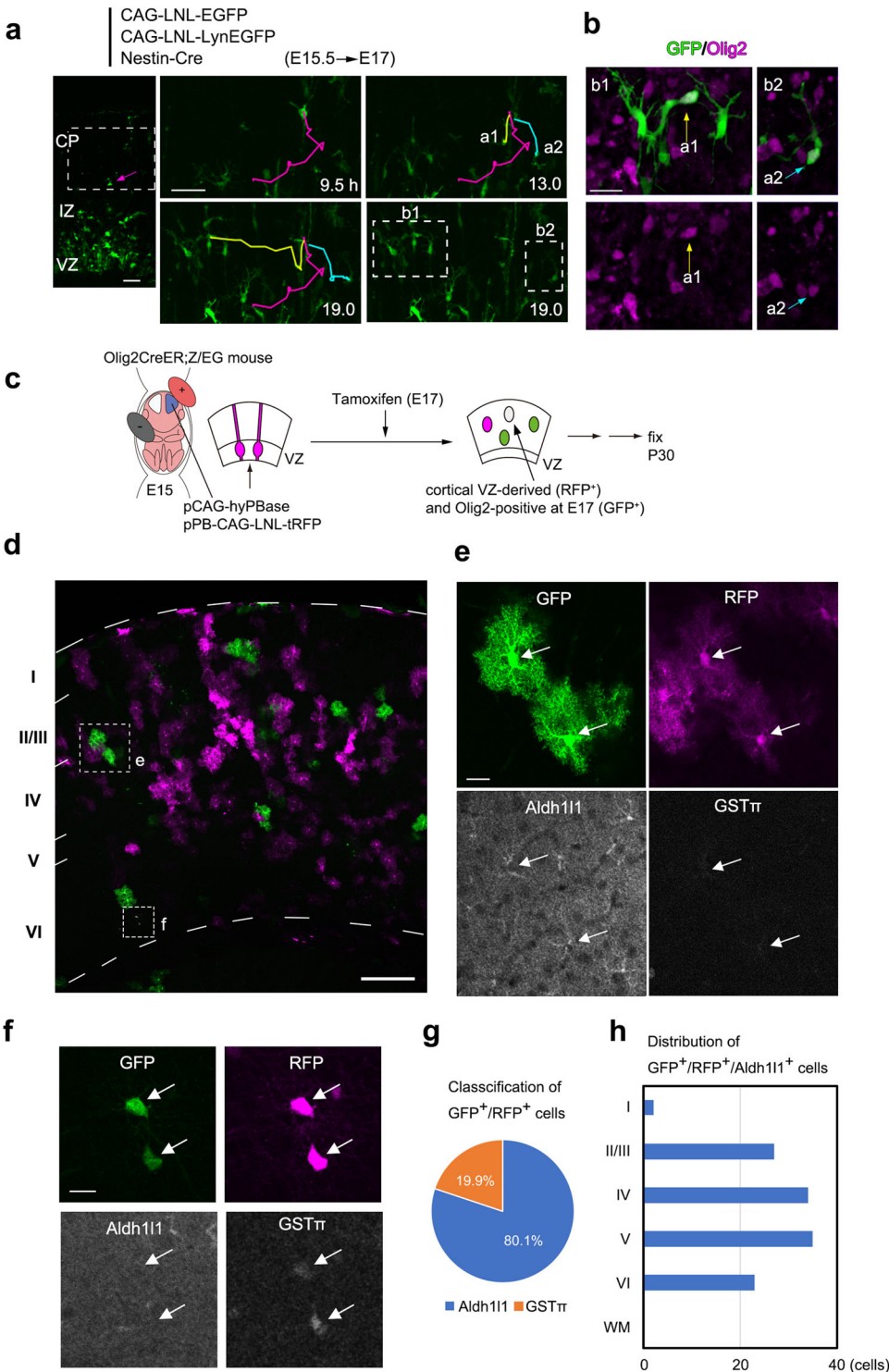

Fig. 6a, Supplementary Movie 10) as we observed in slice culture (Fig. 3d). Beside these migration modes, we observed GFP-positive cells moving in a somal translocation mode, in which the ascending process attached to the basal lamina of the brain surface retracted to bring the cell body toward the brain surface (Supplementary Fig. 6b, Supplementary Movie 13). This migration mode of astrocyte progenitors was proposed and has been widely accepted based on GFAP staining of fixed brains[8]. Here, we obtained direct in vivo evidence of its presence in intact brains. We also performed two-photon imaging at P0 and found that GFP-positive cells were almost stationary at this stage, although a few instances of cells migrating by somal translocation were observed (Supplementary Fig. 6c, Supplementary Movie 14).

Taken together, our data confirmed that, besides somal translocation migration, astrocyte progenitors moved in blood vessel-guided and blood vessel-independent modes in intact living embryos.

### Identification of Cxcr4, Cxcr7 (Ackr3) and Integrin β1 as candidate genes that regulate the blood vessel-guided migration

We observed that the migration pattern of cortical neurons and astrocyte progenitors were completely different from each other (Fig. 1). To address the molecular mechanisms underlying this phenomenon, we used a single cell RNA-seq dataset of E18 mouse whole brains and cell type annotation information[36]. First, we identified 15 clusters in approximately 50,000 cells randomly sampled from the

**Fig. 2 | Cortical VZ-derived Olig2-positive cells differentiate into protoplasmic astrocytes in the cortical gray matter in vivo. a, b** Erratically migration cells express Olig2. **a** E15.5 embryos were electroporated with indicated plasmid and prepared slices two days later. The slice at the beginning of the observation is shown on the left. The trajectory of one cell (indicated by an arrow in the left most panel) is shown with the magenta line. This cell underwent cell division at 9.5–13.0 h and its daughter cells, a1 and a2, also moved in an erratic migration mode (yellow and cyan lines). The last frame of the time-lapse is shown on the bottom right. **b** Immunohistochemistry of the boxed region in the last frame in (**a**) revealed that cells a1 and a2 expressed Olig2. 83% of erratically migrating cells (28 out of 34 cells, in 10 brain slices from three independent cultures) were Olig2 positive. **c** Strategy for fate analysis of cortical VZ-derived Olig2-positive cells in vivo. The cells in Olig2-positive cell lineage were labeled by GFP using *Olig2-CreER;Z/EG* mouse. Tamoxifen was administered at E17 to activate CreER. The cells derived from cortical VZ were labeled by electroporation with a indicated transposon vector system at E15. The double-positive cells were analyzed at P30. **d** An example of brain section containing GFP and turboRFP double-positive cells. **e, f** A quadruple staining for GFP, RFP, Aldh1l1, and GSTπ of the boxed area in (**d**). **e** GFP⁺/turboRFP⁺ cells assuming the morphology of protoplasmic astrocytes (arrows) were positive for Aldh1l1, but negative for GSTπ. **f** Some GFP⁺/turboRFP⁺ cells were Aldh1l1-negative and GSTπ-positive oligodendrocytes. **g** The major population of the cortical VZ-derived cells that expressed Olig2 at E17 differentiated into Aldh1l1⁺ astrocytes (80.1%, 151 cells/18 slices/3 brains). **h** Histogram showing the distribution of GFP⁺/turboRFP⁺ astrocytes in the whole thickness of the pallium. The double-positive astrocytes were only found in the gray matter. Scale bars, 200 μm (**d**), 50 μm (**a**), 20 μm (**b, e, f**). Source data are provided as a Source Data file.

full dataset of 1.3 million cells (Fig. 5a). We then investigated the cell types of each cluster using gene markers (Fig. 5b), and identified clusters 1, 6, and 11 as migrating cortical neurons, glial progenitors, and endothelial cells, respectively. Within cluster 6, OPCs expressing *platelet-derived growth factor receptor A* (*Pdgfra*) and astrocyte progenitors expressing *fatty acid binding protein 7* (*Fabp7*), *vimentin* (*Vim*), *tenascin C* (*Tnc*), and *aldolase C* (*Aldoc*) were included. We excluded *Pdgfra*-expressing cells (OPCs) from cluster 6 and considered the remaining population as astrocyte progenitors (cluster 6e, Fig. 5c). We compared the gene expressions between astrocyte progenitors (cluster 6e) and migrating cortical neurons (cluster 1), and identified 4723 and 2122 differentially expressed genes (DEGs) in astrocyte progenitors and migrating neurons, respectively (Supplementary Fig. 7a, threshold of adjusted p-value <0.01). Gene ontology (GO) terms related to cell division and lipid metabolism were significantly enriched in the astrocyte progenitor-enriched DEGs (Supplementary Fig. 7b), while in the migrating neuron-enriched DEGs GO terms related to neuronal development, axon guidance, and synaptic functions were enriched (Supplementary Fig. 7c). Considering the frequent cell divisions of astrocyte progenitors[9,27] and the central roles of astrocytes in lipid synthesis and delivery to neurons[37], it is reasonable to assign the cluster 6e astrocyte progenitors. To investigate the difference between these two clusters in migration properties, we focused on DEGs related to cell adhesion (GO, biological processes = cell adhesion) and classified them into cadherin superfamily, extracellular matrix proteins, integrins, immunoglobulin superfamily, and others (Supplementary Fig. 7d–g). As a result, various extracellular matrix (ECM) proteins and integrins were identified in the astrocyte progenitor-enriched DEGs, whereas variety of cadherin superfamily and immunoglobulin superfamily genes were found in the migrating neuron-enriched DEGs, highlighting the difference in molecular machinery of these two cell types for migration.

Next, we tried to identify the molecular mechanisms for astrocyte progenitors to be attracted to the blood vessels. We hypothesized that endothelial cells released chemoattractants, for which astrocyte progenitors expressed receptors. We, therefore, explored such molecules in the astrocyte progenitor-enriched DEGs and identified several chemoattractant receptors (Figs. 5c, c1). Additionally, we explored chemotaxis ligands expressed in endothelial cells (cluster 11). We examined these gene lists and identified a chemokine, *Cxcl12*, in endothelial cells and its receptors, *Cxcr4* and *Cxcr7* (*Ackr3*), in astrocyte progenitors. The expression of Cxcr4 in Olig2 positive cells and Cxcl12 in endothelial cells in the developing mouse brains have been already shown[38]. The expression of *Cxcr4* and *Cxcr7* mRNA in astrocyte progenitors (Olig2⁺ cells derived from cortical VZ) was confirmed by in situ hybridization chain reaction (Fig. 5d, e).

### Functional blocking of Cxcr4 and 7 hampered the blood vessel-guided migration

To examine whether Cxcr4 and/or Cxcr7 were involved in blood vessel-guided migration of astrocyte progenitors, we performed in utero gene knockout of each of these receptors in astrocyte progenitors using CRISPR gene editing technology. We generated CRISPR/Cas9 vectors (pX330) targeting *Cxcr4* and *Cxcr7* (Supplementary Fig. 8a–e) and electroporated these vectors, as well as pPB-CAG-EGFP to mark the transfected cells, into E15 brains. The distances from the blood vessels to GFP⁺ cells were assessed at P3 (Fig. 6a, b). The violin plot revealed a prominent peak of vessel-associated cells (less than 5 μm), which was not observed in the plot for DAPI-stained cells (Fig. 6b). The proportion of vessel-associated cells was reduced and the overall distances from vessels were significantly increased (control *vs.* Cxcr4 CR#1, and control *vs.* Cxcr4 CR#2) by the introduction of two different pX330 vectors (CR#1 and CR#2) targeting *Cxcr4*. Although pX330 vectors targeting *Cxcr7* did not induce significant changes in distances from vessels in this assay, co-transfections of two different combinations of pX330 targeting *Cxcr4* and *Cxcr7* increased the distances from vessels more significantly than single transfections of pX330-*Cxcr4* vectors (control *vs.* Cxcr4 CR#2 + Cxcr7 CR#1, and control *vs.* Cxcr4 CR#1 + Cxcr7 CR#2), suggesting complementary actions of Cxcr4 and Cxcr7 in blood vessel-guided migration. During these observations, we did not find altered densities of blood vessels in CRISPR vector-transfected brains, which may possibly have been affected by the migration defect of astrocyte progenitors, considering that they may support the branch formation of blood vessels (Fig.3f, Supplementary Fig. 5). This might be due to the compensation by the untransfected astrocyte progenitors.

Next, to evaluate the effect of gene disruptions on astrocyte progenitor migration along blood vessels, we cultivated endothelial cells forming tubes in vitro, and placed CRISPR vector-transfected astrocyte progenitors onto the resulting endothelial tubes. Time-lapse observations revealed that astrocyte progenitors quickly covered the endothelial tubes and migrated along them, whereas some cells migrated independently of the tubes. We quantified the ratio of cells migrating off the tubes against cell migrating along blood vessels using computer-assisted analysis and found that transfection with pX330 vectors targeting *Cxcr4* or *Cxcr7* significantly increased this ratio (Fig. 6c, Supplementary Movie 15).

To examine the direct effects of CRISPR vectors targeting *Cxcr4* and *Cxcr7* on blood vessel-guided migration in organotypic slice culture, we electroporated these vectors (one *Cxcr4* and two *Cxcr7*-specific CRISPR vectors) into E16 *Flt1-DsRed* mouse brains and performed time-lapse imaging two days later. The migration distances achieved by transfected cells were reduced due to frequent stops and detachment from blood vessels (Fig. 6d, Supplementary Movie 16), suggesting that blood vessel-guided migration was impaired.

### Integrin β1 is involved in Cxcr4-mediated blood vessel-guided migration

Cxcr4 is known to function through integrins in response to Cxcl12[39]. Moreover, integrin genes were enriched in DEGs in astrocyte progenitors (Supplementary Fig. 7e). Thus, we investigated whether integrins were involved in blood vessel-guided migration of astrocyte progenitors. We identified several integrin β genes in DEGs in astrocyte

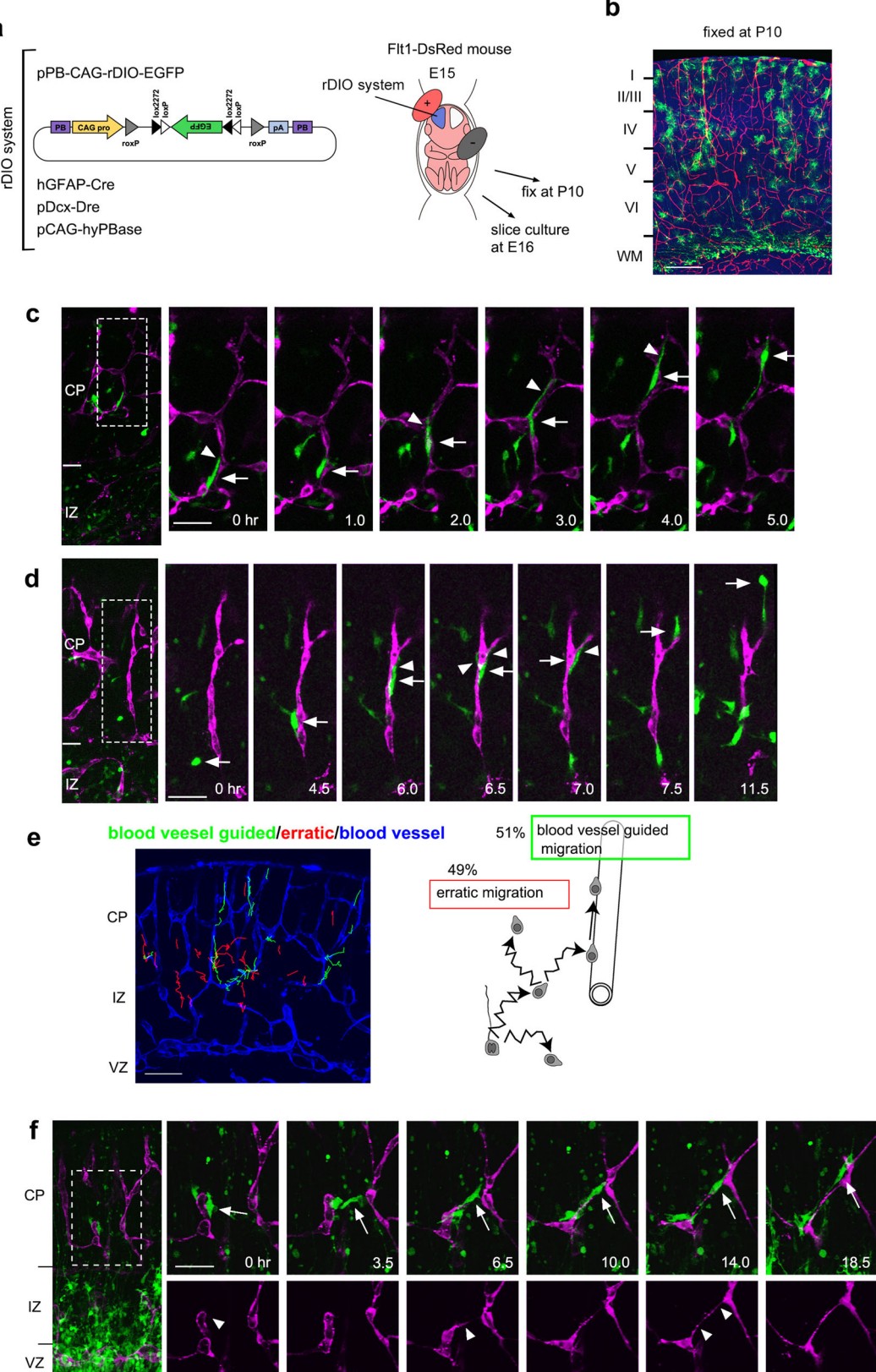

**e** blood veesel guided/erratic/blood vessel

51% blood vessel guided migration

49% erratic migration

progenitors (Fig. 5c, c2, Supplementary Fig. 7e). Among them, Integrin β1 (Itgb1) showed the highest fold-change in expression. To investigate the functional relationship between Cxcr4/7 and Integrin β1 in astrocyte progenitors, we examined the amount of Integrin β1 on Cxcr4/7 KO astrocyte progenitors using a flow cytometer, and found that it was significantly decreased compared to control (Fig. 7a). Then, we constructed knockdown vectors targeting *Itgb1* (miRNA expression vector, Fig. 7b) and examined the effects on blood vessel-guided migration. As we expected, the distances from blood vessels were significantly increased by two distinct knockdown vectors targeting different sequences of Itgb1 (Fig. 7c, d) compared with those measured in controls. On the other hand, cells overexpressing Cxcr4 (Cxcr4-OE),

**Fig. 3 | Blood vessel-guided migration: another migration mode of astrocyte progenitors. a** Plasmid system (rDIO; pPB-CAG-rDIO-EGFP + hGFAP-Cre + pDcx-Dre + pCAG-hyPBase) to label astrocyte progenitors with GFP. E15 *Flt1-DsRed* mouse embryos, which express DsRed in endothelial cells, were electroporated with the rDIO plasmid system and subjected to time-lapse observations next day. **b** The cell-fate of GFP⁺ cells labeled by the rDIO system. The GFP-positive cells were broadly distributed within the CP and differentiated into astrocytes by P10 (*n* = 3 brains). **c** A cell (arrow) protruded a leading process (arrowheads) and migrated rapidly toward the pial surface along blood vessels. See also Supplementary Movie 4. **d** One cell (arrow) attached to a blood vessel (arrowhead, 4.5 h), and migrated along the vessel toward the pial surface, and detached from the blood vessel (11.5 h). See also Supplementary Movie 5. **e** Computer-assisted classification of the astrocyte progenitor migration into vessel-guided or erratic migration. One example of the

analysis using a time-lapse movie of GFP-labeled astrocyte progenitors in a slice from the *Flt1-DsRed* mouse brain electroporated with pCAG-LNL-EGFP + pCAG-LNL-LynEGFP + Nestin-Cre. The blood vessels (DsRed signals, blue), as well as the trajectories of GFP-positive cells migrating on (green) and off (red) the blood vessels are shown (left). Astrocyte progenitors migrate either in the erratic migration mode or the blood vessel-guided migration mode. **f** Astrocyte progenitors support blood vessel branch formation. E15 *Flt1-DsRed* mouse embryos were electroporated with pCAG-LNL-EGFP + pCAG-LNL-LynEGFP + Nestin-Cre and subjected to time-lapse observations next day. The time-lapse images in the boxed regions of the left panels are shown on the right. A cell (arrow) formed a bridge between two adjacent blood vessels, and then one of the blood vessels extended a process (arrowhead) along the astrocyte progenitor and fused to the other. See also Supplementary Movie 6. Scale bars, 200 µm (**b**), 50 µm (**c**, **d**, **f**), 100 µm (**e**).

obtained by introducing pCAG-Myc-Cxcr4, were located closer to vessels than were control cells. This was rescued by simultaneous Itgb1 knockdown (KD#1), and the distance from blood vessels became similar to the Itgb1 KD#1 beyond the control level, indicating that not only overexpressed Cxcr4 but also intrinsic Cxcr4 are depending on Itgb1 for the blood vessel association. These results suggested that astrocyte progenitors contacted blood vessels through the Cxcr4-integrin β1 axis.

## The final position of astrocytes is affected by functional blocking of Cxcr4/7 and Integrin β1

Finally, we assessed whether Cxcr4/7-Integrin β1 defects affected the final localization of astrocytes in the CP. We electroporated with control or CRISPR/Cas9 vectors against *Cxcr4* and *Cxcr7* at E15 and analyzed at P8 (Fig. 8a). In the control experiment, transfected astrocytes were located denser in the superficial CP (bin 1) (Fig. 8b, c) as we observed in Supplementary Fig. 3a and 3c. On the other hand, when we electroporated with two combinations of different CRISPR/Cas9 vectors targeting *Cxcr4* and *Cxcr7*, the amounts of GFP-positive cells in bin 1 were significantly decreased (Fig. 8b, c). The same effect was also observed using a knockdown vector targeting *Cxcr4* (miRNA expression vector, Supplementary Fig. 9). As we expected, knockdown of Itgb1 by using two different vectors (Fig. 8d) also reduced the proportion of transfected astrocytes located in the superficial CP (Fig. 8e, f), and this effect was rescued by introducing an Itgb1-KD#1-resistant Itgb1 expression vector (pPB-CAG-Itgb1-R, Fig. 8f). Taken together, these results indicated that blood vessel-guided migration of astrocyte progenitors depending on Cxcr4/7-Itgb1 axis was necessary for proper astrocyte positioning, especially in the superficial CP.

## Discussion

Astrocytes are among the most abundant cell types in the mammalian brain. During development, astrocytes migrate into the CP to actively participate in the formation of neural circuits, including in synapse formation, maturation, and elimination[7,40], indicating the importance of a spatiotemporally accurate delivery of astrocytes into the cortical gray matter. Despite their importance, the development of astrocytes remains poorly understood. Here, we demonstrated the basic principles underlying astrocyte migration and positioning during cortical development. We showed that astrocyte progenitors derived from prenatal cortical VZ cells were remarkably motile and followed an erratic migration pattern through irregular movements within the IZ and CP or a blood vessel-guided migration process, which was frequently observed in the superficial CP. They eventually differentiated into protoplasmic astrocytes extending highly branched fine processes from the cell body. On the other hand, the astrocytes derived from postnatal VZ cells remained in the white matter or in deep positions in the cortical gray matter and differentiated into GFAP-strongly positive fibrous astrocytes. We also showed that Cxcr4 and Cxcr7 were involved in blood vessel-guided migration probably by activating integrin β1. Functional blocking of

Cxcr4 and Cxcr7, as well as integrin β1, reduced the distributions of astrocytes in the superficial CP, suggesting that blood vessel-guided migration was important for efficient delivery of astrocytes near the brain surface.

In erratic migration, proliferative astrocyte progenitors moved in various directions. In accordance with our observations, clonal dispersions of astrocytes within the cortical gray matter and the intermixing of clones were previously described[9]. We assume that this random dispersion facilitates the even distribution of astrocytes throughout the CP. Protoplasmic astrocytes form mutually exclusive territories by extending highly branched processes. To achieve this, astrocytes need to be spread apart from each other. Similar cellular behaviors are known in interneuron migration. Interneurons migrate into the cortical marginal zone (MZ) from the medial ganglionic eminence, and then move in various directions tangentially within the MZ before diving into the CP[41,42]. This process of multidirectional interneuron migration is believed to contribute to their even dispersion throughout the cortex. The erratic migration of astrocyte progenitors might have a similar role of dispersing astrocytes evenly in the cortex.

A blood vessel-guided migration has been reported for other cell types, such as OPCs[29,38]. Tsai et al. demonstrated that OPCs extend a short leading process, migrate on blood vessels, and sometimes jump between vessels. These observations are very similar to what we saw for astrocyte progenitors. Both glial progenitors express Olig2, but they are easily distinguished by their specific markers [platelet-derived growth factor receptor A (PDGFRα) and neural/glial antigen 2 (NG2) for OPCs, and Aldh1l1 for astrocyte progenitors]. We demonstrated that GFP⁺ cells in Aldh1l1-GFP mice and cells labeled with rDIO, which is specific for astrocyte progenitors, were highly associated to and migrated on blood vessels, indicating that cells migrating on blood vessels we observed here were astrocyte progenitors and not OPCs. OPCs and astrocyte progenitors are distinct populations; however, their cell morphology and migratory behaviors are strikingly similar. In fact, both populations are attracted to blood vessels through Cxcr4 signaling[38], suggesting common molecular mechanisms for blood vessel-guided migration. Glioma cells arise from astrocytes or oligodendrocytes and express the undifferentiated glial cell marker, Olig2. They infiltrate the brain along blood vessels[43] relying on the Cxcr4 signaling[44,45]. This cellular behavior appears to be a recapitulation of blood vessel-guided migration of astrocyte progenitors during brain development.

The switching mechanisms between erratic and vessel-guided migrations are still obscure. During the cortical development, astrocyte progenitors migrate from the ventricular zone to the cortical plate, and we observed that, in the superficial cortical plate, vessel-guided migration became dominant against erratic migration, indicating that some determinant for vessel-guided migration might be upregulated as the differentiation or maturation process of astrocyte progenitors proceeds. In this study, we demonstrated that Cxcr4/7 and Integrin β1 are key factors for vessel-guided migration. Hence, the expression or activation of these molecules or their regulatory factors

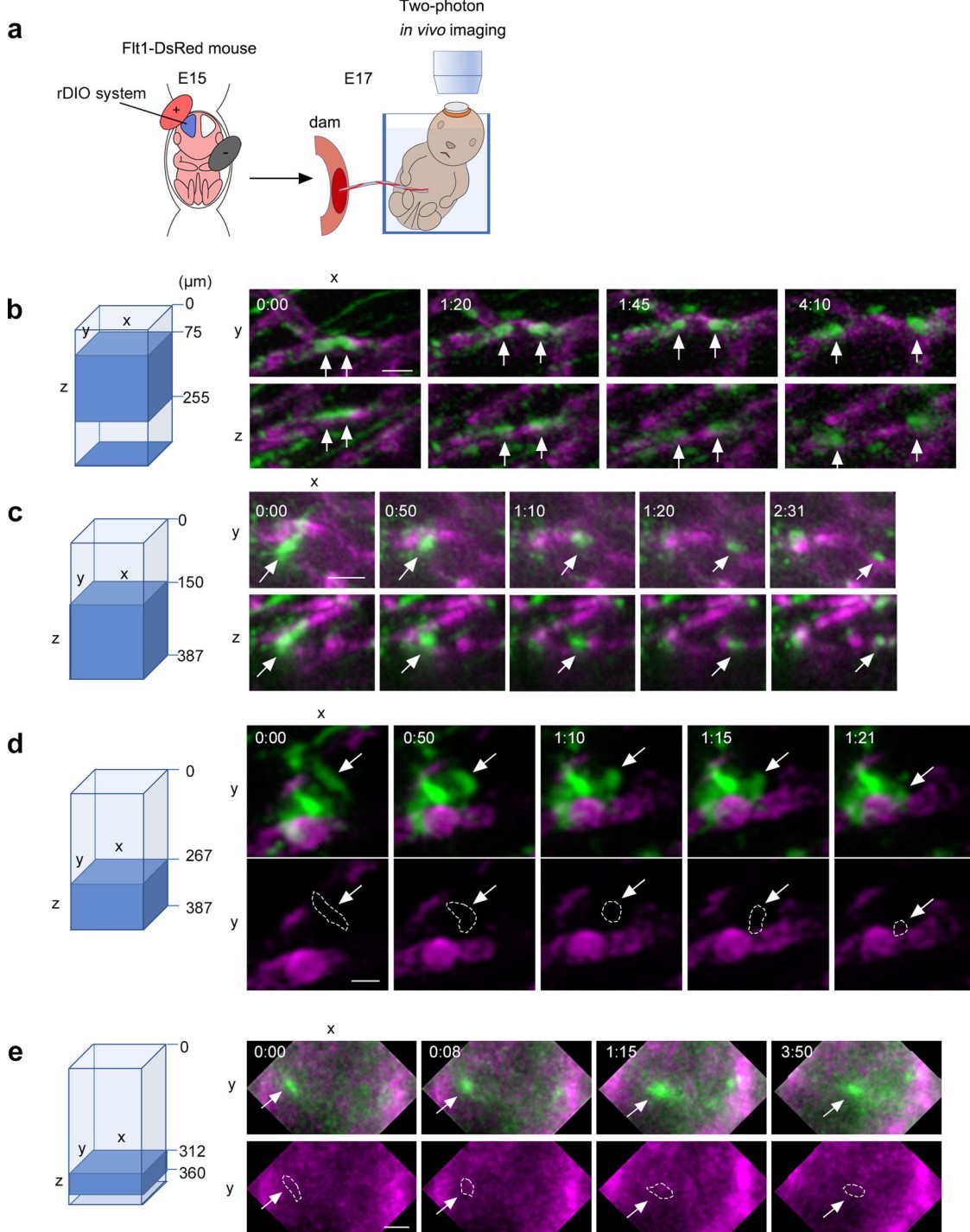

**Fig. 4 | Two-photon in vivo imaging of blood vessel-guided migration and blood vessel-independent migration. a** Schematic illustration of two-photon imaging of living embryos we performed. We introduced the rDIO system into *Flt1-DsRed* mice at E15 and performed imaging 2 days later. (**b**–**e**) Reconstructed two-photon images. The range of Z axis to reconstruct the image is shown on the left (blue region, the scale of depth from the brain surface is shown on the right side of the box). The time stamps are shown in h:mm format. (**b**, **c**) Two examples of in vivo imaging showing blood vessel-guided migration of GFP-positive cells (arrows). The upper and lower panels represent xy and xz planes of the same 3D space, respectively. The movie versions of **b**, **c** are provided as Supplementary Movie 8 and 9, respectively. (**d**, **e**) Two examples of in vivo imaging showing blood vessel independent migration of GFP-positive cells (arrows). The merged images of GFP and DsRed, as well as DsRed single channel, are shown in the upper and lower panels, respectively. The broken lines in the lower panels represent the outlines of GFP-positive cells. The movie versions of **d**, **e** are provided as Supplementary Movie 11 and 12, respectively. Scale bars, 20 μm.

might be changed as they migrate toward the brain surface, which should be addressed in the future study.

Our observations of blood vessel-guided migration suggested that astrocyte progenitors might enhance branch formation of blood vessels. This observation further indicates the physiological importance of blood vessel-guided migration, beyond the efficient delivery of astrocytes on the brain surface. In accordance with this, a previous study demonstrated that the inhibition of astrocyte production during the early postnatal period result in severe reduction of the density and branching frequency of cortical blood vessels[46]. Branch formation is

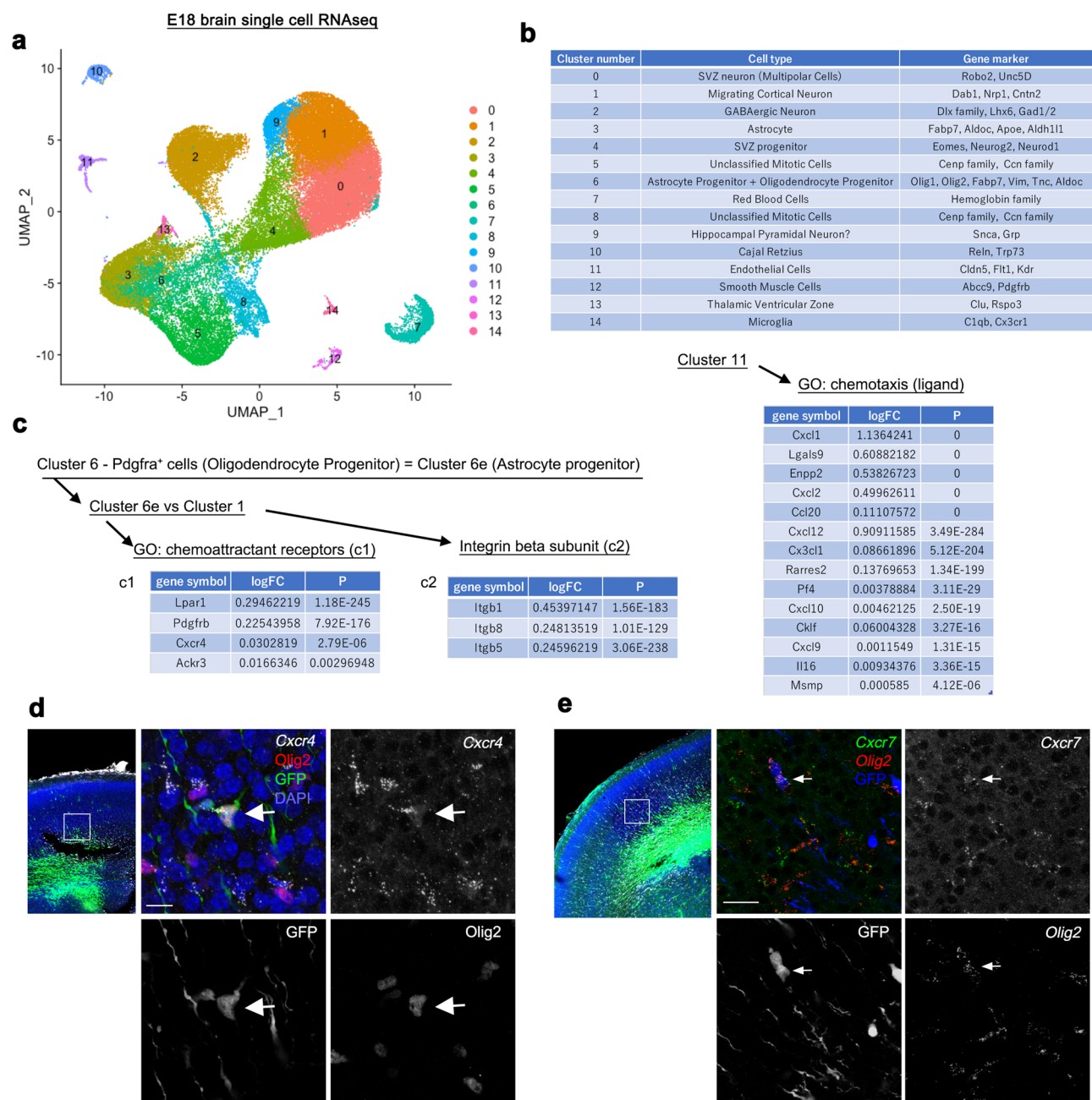

**Fig. 5 | Identification of Cxcr4 and Cxcr7 (Ackr3) and Integrin β1 as candidate molecules that regulate the blood vessel-guided migration. a** UMAP (uniform manifold approximation and projection) visualization of approximately 50,000 single cell RNA-seq data randomly sampled from the full dataset (1.3 million cells) of E18 mouse whole brains. 15 clusters were identified. **b** Annotation of cell types for the 15 clusters based on gene markers as indicated. Clusters 1, 6 and 11 represent migrating cortical neurons, glial progenitors (astrocyte progenitors and oligodendrocyte progenitors), and endothelial cells, respectively. **c** Identification of astrocyte-specific chemoattractant receptors (c1) and their candidate ligands secreted from endothelial cells (right). We identified Cxcr4 and Cxcr7 (Ackr3), which are receptors for Cxcl12 secreted from endothelial cells, as candidate molecules that regulate the blood vessel-guided migration. We also identified

integrin β family expressed in astrocyte progenitors (c2). logFC, log fold-change of the average expression of astrocyte progenitors versus Cluster 1 (c1, c2), or that of endothelial cells versus all cells (right). P, adjusted *p*-value. We applied two-sided Wilcoxon Rank Sum test with adjustments for multiple comparisons based on bonferroni correction. (**d**, **e**) Expression of Cxcr4 and Cxcr7 in cortical VZ-derived Olig2-positive cells. E15 mouse embryos were electroporated with PB-CAG-EGFP and fixed 3 days later (**d**, **e**, left panel). Some Olig2 (detected by immunohistochemistry (**d**) or HCR (**e**)) and GFP double-positive cells indicated by arrows were positive for Cxcr4 (HCR, 28.29 ± 3.36%, 131 cells/4 brains, mean ± SEM) and Cxcr7 (HCR, 35.28 ± 4.79%, 141 cells/4 brains, mean ± SEM). Scale bars, 10 μm (**d**), 20 μm (**e**).

also supported by macrophages/microglia in the hind brain and retina[47]. Interestingly, microglial cell behaviors observed upon branch formation of blood vessels strikingly resemble those of astrocyte progenitors; that is, microglia form a bridge between two vessel sprouts, which extend along the bridge to fuse together. This suggests the existence of shared mechanisms triggered by microglia and astrocyte

progenitors to support branch formation of blood vessels. Further analysis will reveal the role of astrocyte progenitors in angiogenesis.

The most widely accepted model of astrocyte migration is the somal translocation mode, which was proposed based on the histological analysis of Golgi staining[48] and the GFAP+ transforming radial glia in ferret and human fetuses[8,49]. Here, we observed this migration

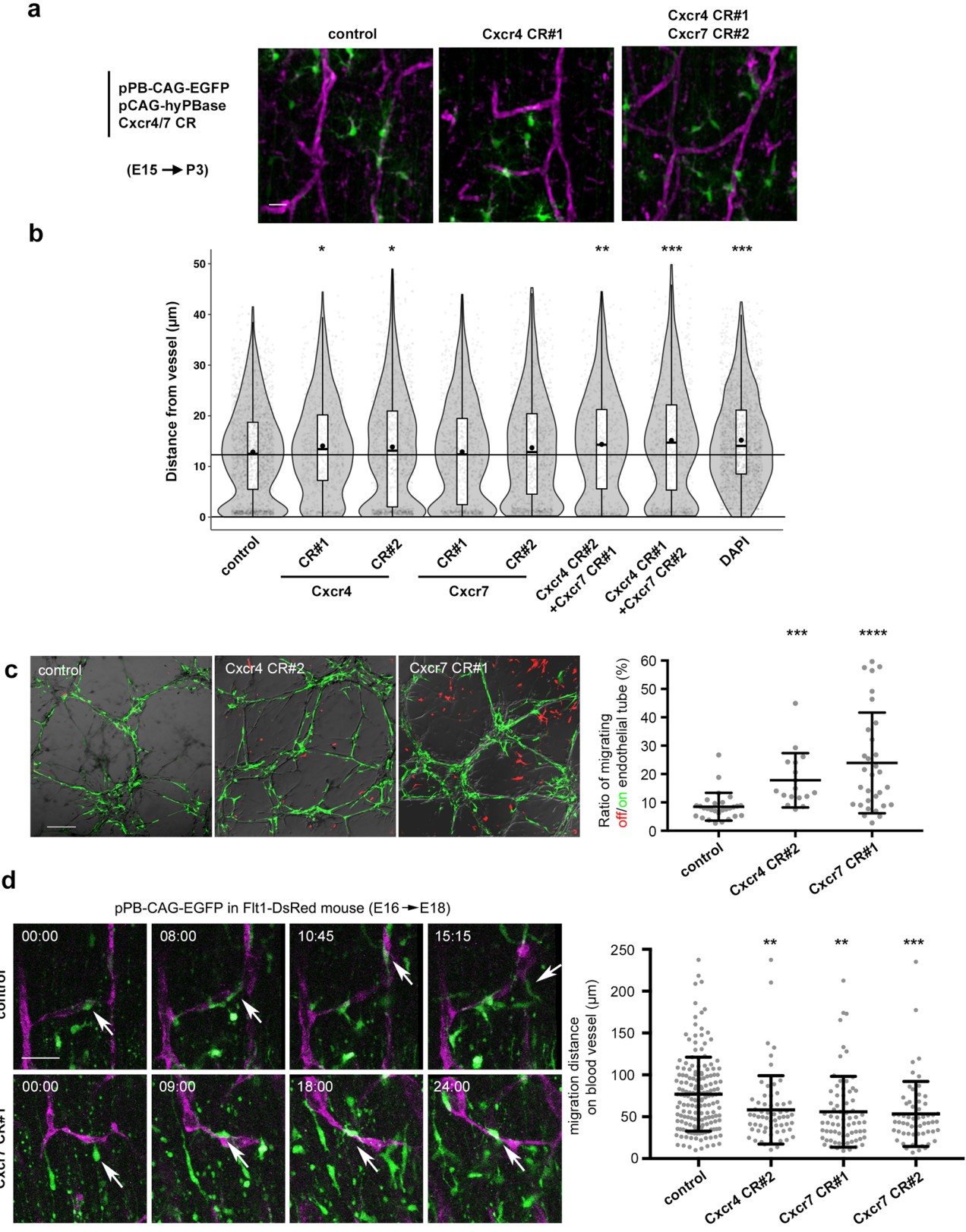

mode near the brain surface by two-photon live imaging. We hypothesize that the cells using somal translocation likely settle beneath the pial surface and differentiate into GFAP-strongly positive layer I astrocytes. On the other hand, the highly motile astrocyte progenitors that follow erratic migration and blood vessel-guided migration patterns were only weakly positive for GFAP, and entered the cortical gray matter to differentiate into protoplasmic astrocytes. These highly

motile astrocyte progenitors were produced from VZ cells at restricted prenatal stages (E15–E17 in mice), whereas astrocytes derived from postnatal VZ remained around the white matter and differentiated into GFAP-positive fibrous astrocytes. Therefore, we propose that both basic astrocyte types, namely protoplasmic and fibrous astrocytes, are produced at different timing from VZ cells. In the future, it will be important to determine whether this timing difference is

**Fig. 6 | Functional blocking of Cxcr4 and 7 hampered the blood vessel-guided migration. a** CRISPR vectors against Cxcr4/7 disrupt the association of astrocyte progenitors with blood vessels. **b** Distances from blood vessels to transfected astrocyte progenitors (GFP⁺/Aldh1l1⁺). Cxcr4 CR#1 and #2 significantly increased the distances from the blood vessels, which were enhanced by adding Cxcr7 CR#1 or #2 (two-sided Dunnett's test, 1767 cells/14 brains control, 725 cells/6 brains Cxcr4 CR#1, 1166 cells/7 brains Cxcr4 CR#2, 936 cells/8 brains Cxcr7 CR#1, 1194 cells/16 brains Cxcr7 CR#2, 557 cells/10 brains Cxcr4 CR#2 + Cxcr7 CR#1, 808 cells/ 7 brains Cxcr4 CR#1 + Cxcr7 CR#2, 1921 cells/7 brains, control *vs.* Cxcr4 CR#1, $P = 0.0345$; control *vs.* Cxcr4 CR#2, $P = 0.0448$; control *vs.* Cxcr4 CR#2 + Cxcr7 CR#1, $P = 0.0106$; control *vs.* Cxcr4 CR#1 + Cxcr7 CR#2, $P < 0.0001$). For detailed information of box plots, see "Statistical analysis" section in Methods. **c** Glial progenitors transfected with CRISPR vectors were cultured on endothelial tubes, and were pseudo-colored in green and red for those migrating on and off the

endothelial tubes, respectively (left). Ratio of cells migrating off versus on the endothelial tubes in each objective field (right, 28 control, 17 Cxcr4 CR#2, and 32 Cxcr7 CR#1 objective fields/3 experiments, Kruskal-Wallis test ($P < 0.001$) followed by two-sided Dunn's multiple comparisons test, control *vs.* Cxcr4 CR#2, $P = 0.005$; control *vs.* Cxcr7 CR#1, $P < 0001$, no significant difference between Cxcr4 CR#2 and Cxcr7 CR#1, $P = 0.999$). See also Supplementary Movie 15. **d** *Flt1-DsRed* mouse embryos were electroporated with CRISPR vectors and employed to time-lapse observations (left panels). CRISPR vectors reduced the migration distance on blood vessels (right panel, 169 control, 63 Cxcr4 CR#2, 72 Cxcr7 CR#1, and 59 Cxcr7 CR#2 cells/3 experiments, two-sided Dunnett's multiple comparisons test, control *vs.* Cxcr4 CR#2, $P = 0.0089$, control *vs.* Cxcr7 CR#1, $P = 0.0015$, and control *vs.* Cxcr7 CR#2, $P = 0.0008$). See also Supplementary Movie 16. Scale bars, 20 μm (**a**), 200 μm (**c**), 50 μm (**d**). Horizontal bars in scatter plots (**c**, **d**) represent mean ± SD. Source data are provided as a Source Data file.

caused by developmental changes in cell intrinsic mechanisms or by extrinsic effects.

Recent studies have demonstrated the layer-specific diversity of cortical astrocytes[50–52]; however, the mechanisms of this diversification are not yet clear. We observed that astrocyte progenitors derived from the E15 VZ were localized more superficially than those deriving from the E17 VZ (Supplementary Fig. 3a, c). Moreover, fibrous astrocytes in the white matter were generated postnatally, indicating that astrocytes tend to be aligned in an outside-in manner during cortical development, although this trend was not as clear as the birthdate-dependent positioning of cortical neurons. This birthdate-dependent positioning of astrocytes might partially underlie the layer-dependent differences among astrocytes. Another related difference between superficial and deep cortices is the migration properties. Blood vessel-guided migration was more often observed in the superficial cortex, whereas erratic migration was dominant in the deep cortex. Astrocyte progenitors leaving the VZ relatively early might have different affinity to blood vessels compared with that of astrocytes leaving the VZ later during development.

The relationship between astrocyte dysfunction and neurodevelopmental and psychiatric disorders has been recently investigated[53–55]. It is assumed that developmental abnormalities of astrocytes are involved in disease pathogeneses, although the underlying mechanisms have yet to be uncovered. Astrocytes migrate into the cortical gray matter just at the time when neuronal synaptogenesis begins and play essential roles in this process. In the present study, we clarified the migration properties of astrocyte progenitors destined to the cortical gray matter. When these migratory mechanisms are disrupted by genetic or environmental factors, synaptogenesis of neurons is likely to be affected because of the aberrant positioning of astrocytes. Thus, the basic principles of astrocyte development reported here might contribute to understanding the pathogenesis of neurodevelopmental disorders in the future.

## Methods

### Mice

Timed pregnant ICR mice (Japan SLC, Shizuoka, Japan) were purchased. *Olig2-CreER*[24] and *Z/EG*[25] mice (both 8–24 weeks) were maintained and bred at Keio University School of Medicine. *Flt1-DsRed* mice[35] were maintained by crossing heterozygote males with wild type ICR females (both 8–24 weeks) at Institute for Developmental Research. Aldh1l1-GFP mice[56] (MMRRC, Stock #011015) were obtained from the University of California at Davis. Heterozygous male mice were mated with C57B/6 J females (both 8–24 weeks) at Keio University School of Medicine. The stages of manipulation and sampling were indicated in the corresponding Figure legends (embryonic day 15 to postnatal day 30). All mice were housed at 22–24 °C with 40–60% humidity under the 12 h light/12 h dark cycle. Noon on the date that a vaginal plug was observed was considered as embryonic day 0.5 (E0.5). All protocols for animal handling and treatments conducted in

Institute for Developmental Research were approved by the Animal Care and Use Committee of Institute for Developmental Research, Aichi Developmental Disability Center (approval number: 2019-013). Those in Keio University were approved by the Keio University Institutional Animal Care and Use Committee in accordance with the Institutional Guidelines on Animal Experimentation at Keio University (approval number: A2021-030).

### Immunohistochemistry

Embryos or postnatal mice were perfused with phosphate buffered saline (PBS) followed by 4% paraformaldehyde (PFA) in PBS. The brains were dissected out and were soaked in 4% PFA for at least 2 h. After washing with PBS, the brains were sectioned coronally at 100 μm using a vibrating microtome (VT1000, Leica or HM650-V, Zeiss or Thermo Scientific). Brain slices obtained from around the level of the interventricular foramen were used. The sections were placed onto MAS-coat slides (Matsunami Glass) and treated with HistoVT One (Nacalai Tesque, Cat# 06380-05) at 70 °C for 20 min. After washing with PBS containing 0.05% Tween (PBST), the sections were blocked with 4% BSA in PBST and treated with primary antibodies. The primary antibodies used in this study were mouse anti-GFAP (Sigma-Aldrich, Cat# 3893, clone G-A-5, 1:500), goat anti-Olig2 (R&D, AF2418, 1:300), rabbit and chicken anti-GFP (MBL, Cat# 598, 1:1000; Aves, Cat# GFP-1010, 1:3000), rabbit anti-Aldh1l1 (Abcam, Cat# ab87117, 1:1000), goat anti-GSTπ (LifeSpan Biosciences, Cat# LS-B2376, 1:300) and goat anti-S100β (R&D, Cat# AF1820, 1:200). After washing with PBST three times, the sections were stained with secondary antibodies; Alexa Fluor 488-conjugated anti-Chicken IgY (donkey, Jackson ImmunoResearch Laboratories 703-545-155, 1:1000), Alexa Fluor 555-conjugated anti-rabbit IgG (donkey, Invitrogen, A31572, 1:1000), Alexa Fluor 647-conjugated anti-mouse IgG (goat, Invitrogen, A21236, 1:1000), Alexa Fluor 647-conjugated anti-rabbit IgG (goat, Invitrogen, A21245, 1:1000), Alexa Fluor 647-conjugated anti-goat IgG (donkey, Jackson ImmunoResearch Laboratories 705-605-147, 1:1000), Alexa Fluor 405-conjugated anti-rabbit IgG (goat, Invitrogen, A31556, 1:1000). Unless otherwise 405-conjugated secondary antibody were used, sections were counterstained with DAPI (0.2 μg/ml, Cat# D9542, Sigma-Aldrich). In case of the quadruple staining shown in Fig. 2e, f, Aldh1l1 (rabbit) and GSTπ (goat) were stained with Alexa Fluor 405-conjugated and 647-conjugated secondary antibodies (see above), respectively. The blood vessels were labeled with isolectin B4 conjugated with biotin (Sigma-Aldrich, L2140, 1 μg/ml) and streptavidin-Alexa Fluor 555 (Life Technologies, Cat# S32355, 1:1000). Fluorescent images were captured with a confocal laser microscope (FV-1000; Olympus, or LSM-880; Zeiss).

### Construction of plasmids

pCAG-LNL-GFP was constructed by exchanging the DsRed cDNA of pCALNL-DsRed, a gift from Connie Cepko (Addgene plasmid # 13769)[57],

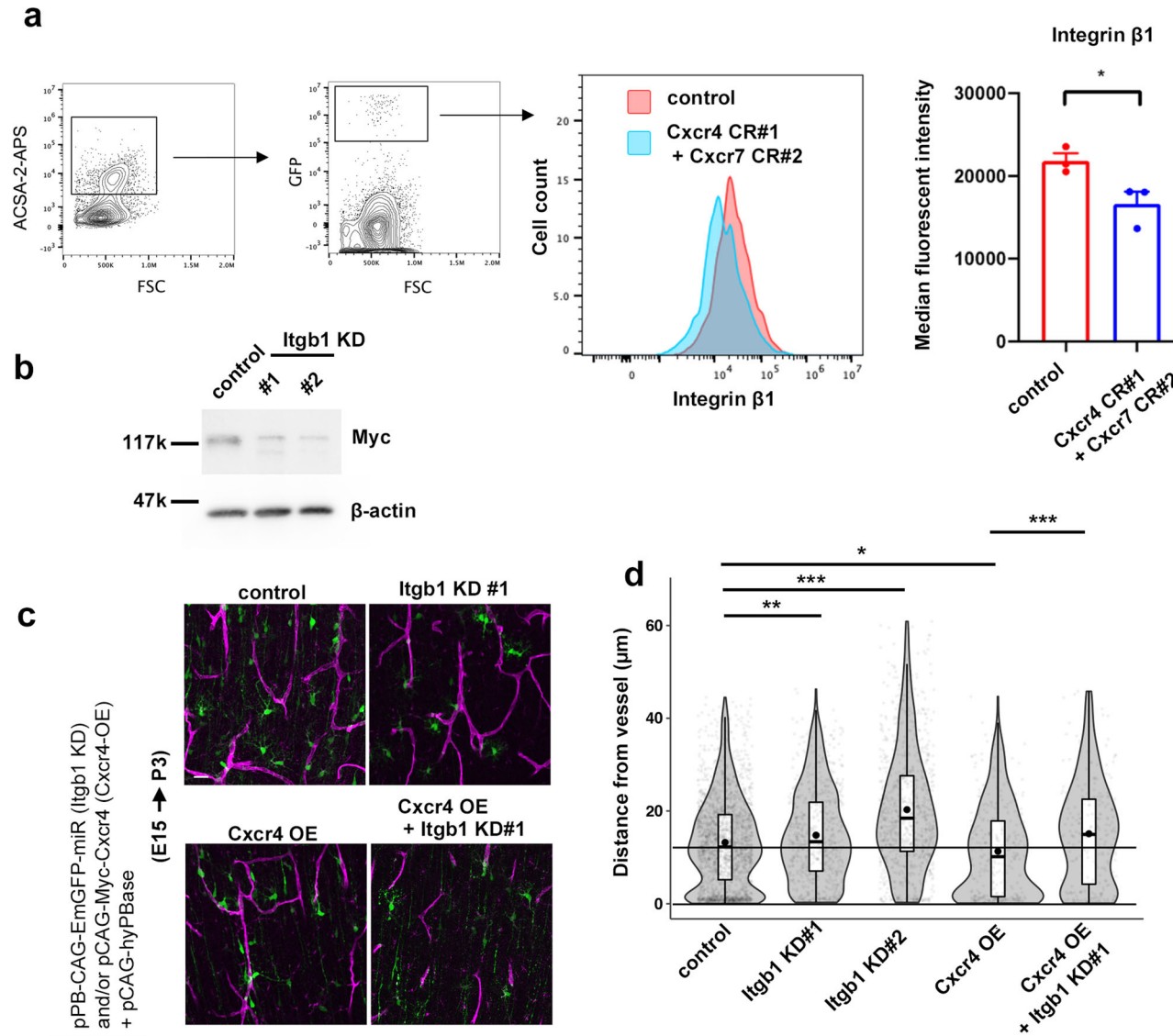

**Fig. 7 | Integrin β1 is involved in Cxcr4-mediated blood vessel-guided migration. a** Simultaneous introduction of Cxcr4 and Cxcr7 CRISPR vectors reduced the amount of Integrin β1 on astrocyte progenitors (3 brains/group, two-sided unpaired Student's t test, $P = 0.0383$, the data in the right panel are shown as mean ± SEM). CRISPR vectors were electroporated at E15, and the amount of Integrin β1 on the GFP and ACSA-2 double positive population at P3 was examined using a flow cytometer. The left dot plot shows the fluorescence intensity of the ACSA-2 (y-axis) versus the intensity of forward scatter (FSC; x-axis). The right dot plot shows the fluorescence intensity of the GFP (transfected cells, y-axis) versus the intensity of forward scatter (FSC; x-axis). Histogram shows the number of events (cells) counted (y-axis) versus the fluorescence intensity of integrin β1 (x-axis) (red, ACSA-2[+], GFP[+] cells electroporated with a control vector; blue, ACSA-2[+], GFP[+] cells co-electroporated with Cxcr4 CR#1 and Cxcr7 CR#2). **b** Knockdown

efficiency of two independent vectors (KD#1 and KD#2) for Itgb1 ($n = 4$ independent transfections to Neuro2a cells). **c** E15 embryos were electroporated with knockdown vectors and/or Cxcr4 expression vector and fixed at P3. Scale bar, 20 μm. **d** Distances from blood vessels to transfected astrocyte progenitors (GFP[+]/Aldh1l1[+]). Both KD#1 and KD#2 significantly increased the distances from blood vessels. Cxcr4 overexpression enhanced the association with blood vessels, while this effect could be canceled by adding KD#1 (two-sided Tukey-Kramer tests, 1929 cells/18 brains control, 691 cells/8 brains Itgb1 KD#1, 844 cells/9 brains Itgb1 KD#2, 325 cells/13 cells Cxcr4 OE, and 245 cells/7 brains Cxcr4 OE + Itgb1 KD#1, control *vs.* KD#1, $P = 0.01406$; control *vs.* KD#2, $P = 0.00321$; control *vs.* Cxcr4-OE, $P = 0.02445$; Cxcr4-OE *vs.* Cxcr4-OE + Itgb1 KD#1, $P < 0.001$). For detailed information of box plots, see "Statistical analysis" section in Methods. *$p < 0.05$, **$P < 0.01$, ***$P < 0.001$. Source data are provided as a Source Data file.

with enhanced green fluorescent protein (EGFP) cDNA (pEGFP-N1, Clontech). pCAG-EGFP and pCAG-kikGR were constructed by inserting EGFP (Clontech), LynEGFP (a gift from A. Miyawaki)[23] or kikGR cDNA (a gift from A. Miyawaki)[14] into a modified pCAGGS vector[58], respectively. Tα1-EGFP was constructed by inserting the EGFP cDNA into the downstream of the Tα1 promoter in plasmid 253 (a gift from P. Barker and F.D.Miller)[59]. Nestin-Cre was constructed by inserting M-Cre (modified Cre recombinase whose codon usage are optimized for mammalian cells) from pCXN-Cre plasmid[60], a gift from Masaru Okabe, into Xh5 plasmid[61], a gift from Urban Lendahl, under the Nestin promoter. The Tol2 transposon system (a transposase expression vector,

pCAGGS-T2TP; a transposon vector, pT2K-CAGGS-EGFP) was a gift from K. Kawakami and Y. Takahashi[62–64]. The PiggyBac transposon vector system was provided from Sanger Institute (a transposase expression vector, pCMV-hyPBase; a transposon vector pPB-CAG.EBNXN). To ensure higher expression of hyPBase (PiggyBac transposase) in the brain, we took out the hyPBase cDNA from pCMV-hyPBase and subcloned it into the downstream of the CAG promoter of the modified pCAGGS vector. To make EGFP expression transposon vector (pPB-CAG-EGFP), we inserted EGFP cDNA (Clontech) into the cloning site of pPB-CAG.EBNXN. To make Cre dependent EGFP expression PiggyBac vector (pPB-CAG-LNL-EGFP), we inserted a

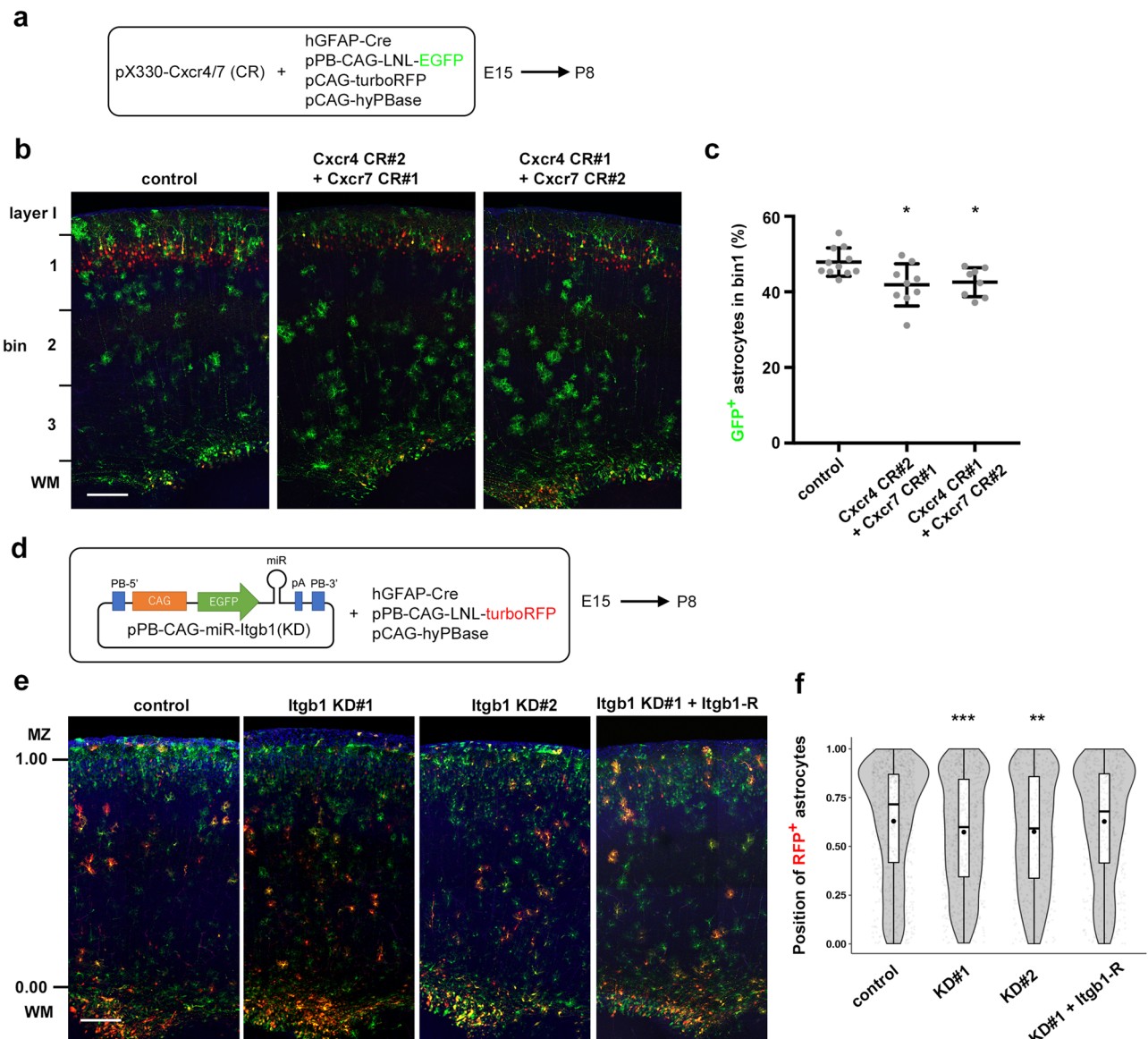

**Fig. 8 | The final distribution of astrocytes is affected by functional blocking of the Cxcr4/7-Integrin β1 signaling axis. a–c** CRISPR vectors for Cxcr4 and 7 disrupted the positioning of astrocytes in the CP at P8. **a** E15 brains were electroporated with the indicated combinations of CRISPR vectors for Cxcr4 and 7, and were fixed at P8. **b** Resulting brains. Astrocytes and neurons were labeled with GFP and RFP, respectively. **c** The astrocytes electroporated with CRISPR-vectors for Cxcr4 and Cxcr7 were reduced in bin 1 (horizontal bars represent mean ± SD, two-sided Dunnett's multiple comparison test, control (11 brains) vs. Cxcr4 CR#2 + Cxcr7 CR#1 (9 brains), $P = 0.0108$; control vs. Cxcr4 CR#1 + Cxcr7 CR#2 (8 brains), $P = 0.0298$ in bin 1). **d–f** Knockdown of Itgb1 also disrupted the distribution of astrocytes within the CP. **d** E15 embryos were electroporated with indicated

plasmids, and fixed at P8. In the rescue experiment, KD#1 resistant Itgb1 expressing vector (pPB-CAG-Itgb1-R) was added in the above plasmid mixture. **e** RFP labels a subset of transfected astrocytes. **f** The radial distributions of RFP$^+$/Aldh1l1$^+$ cells are shown in violin plots. Statistical analysis revealed significant shift of Itgb1 KD cells toward deep CP, which could be rescued by pPB-CAG-Itgb1-R (two-sided Dunnett's multiple comparison test, 891 cells/8 brains control, 764 cells/11 brains Itgb1 KD#1, 497 cells/8 brains Itgb1KD#2, 521 cells/7 brains Itgb1 KD#1 + Itgb1-R, control vs. Itgb1 KD#1, $P = 0.000244$, control vs. Itgb1 KD#2, $P = 0.002399$, control vs. Itgb1 KD#1 + Itgb1-R, $P = 0.999476$). For detailed information of box plots, see "Statistical analysis" section in Methods. $*p < 0.05$, $**P < 0.01$, $***P < 0.001$. Scale bars, 200 μm. Source data are provided as a Source Data file.

loxP-Neo-loxP (LNL) cassette from pCALNL-DsRed into the downstream of the CAG promoter of pPB-CAG-EGFP. The RFP version of Cre dependent PiggyBac vector (pPB-CAG-LNL-turboRFP) was constructed by exchanging the EGFP coding region with turboRFP cDNA (Evrogen). To perform plasmid-based gene editing, we used pX330-U6-Chimeric_BB-CBh-hSpCas9 (pX330, a gift from Feng Zhang, Addgene plasmid # 42230)[65]. A pair of oligonucleotides as listed below was annealed and ligated into BbsI sites of short guide RNA (sgRNA) scaffold of pX330. Regarding the construction of gene-editing dependent puromycin resistance gene expression vector, a pair of annealed oligonucleotides as shown below was inserted into the SacI and AatII sites of pMB1610_pRR-Puro (pRR-Puro), a gift from M. Buhler

(Addgene plasmid # 65853)[66]. We then amplified the split puromycin N-acetyltransferase coding sequence including the target sequence in between, and inserted the PCR products into the downstream of CAG promoter of pPB-CAG.EBNXN vector. To evaluate gene-editing efficiency of CRISPR vectors against Cxcr7 (Cxcr7 CR#1 and CR#2), pCAG-EGxxFP-Cxcr7 was constructed. To obtain the 543 bp DNA fragment which contains target sequences of Cxcr7 CR#1 and CR#2, we performed PCR using a primer set (5'-CAACCACtgaggatcctgatttcacacct aggacacagcca, and 5'-TGCCgatatcgaattccaagatgtagcagtgcgtgtcgtag) and a genomic DNA from ICR mouse as a template. The PCR product was inserted into EcoRI/BamHI sites of pCAG-EGxxFP, a gift from Masahito Ikawa (Addgene plasmid #50716)[67], using Gibson

assembly[68]. hGFAP-Cre was a gift from Albee Messing (Addgene plasmid # 40591)[20]. To construct pPB-CAG-rDIO-EGFP plasmid, annealed synthetic DNA (5'-cctcgagTAACTTTAAATAATGCCAATTATTTAAAGTT AggatccgaattcgatatcgcggccgcaagcttTAACTTTAAATAATGCCAATTATT TAAAGTTAatcgatctcgagggtac, and 5'-cctcgagatcgatTAACTTTAAAT AATTGGCATTATTTAAAGTTAaagcttgcggccgcgatatcgaattcggatccTAAC TTTAAATAATTGGCATTATTTAAAGTTActcgaggagct), which contains the rox sites (capital letters), was inserted into SacI and KpnI site of pBluescript II KS(-) (pBS-roxP). Sense and antisense synthetic DNA of the DIO cassette sequence of double-floxed inverse ChR2-EYFP vector[32], as well as the GFP cDNA from pEGFP-N1 (Clontech) in the reverse direction, were inserted into the AAV-CAG backbone obtained from AAV-CAG-GFP, a gift from Karel Svoboda (Addgene plasmid # 28014; http://n2t.net/addgene:28014; RRID: Addgene_28014)[69] (AAV-CAG-DIO-EGFP). The BamHI/HindIII fragment from AAV-CAG-DIO-EGFP containing DIO-EGFP cassette was inserted into the same restriction enzyme sites of pBS-roxP (pBS-roxed-DIO-EGFP), and then XhoI/ClaI fragment of pBS-roxP containing roxed DIO-EGFP cassette was inserted into the same restriction enzyme sites of pPB-CAG-EGNXN. To construct pDcx-Dre, the *Doublecortin* (DCX) promoter from Dcx4kb-EGFP[70], a gift from Q. Lu (Beckman Research Institute of the City of Hope), was exchanged with CAG promoter of pCAG-NLS-HA-Dre[34], a gift from Pawel Pelczar (Addgene plasmid #51272). For the knockdown of Cxcr4 and Itgb1 using a miRNA system (Invitrogen), the targeting sequence of Cxcr4 and Itgb1 (the sequences of oligonucleotides were shown below) as well as lacZ (control, miRneg, the target sequence is as shown in the kit manual) were inserted into a modified Block-iT Pol II miR RNAi expression vector (Invitrogen)[71,72]. The transcription units containing EmGFP and miRNA sequence for Cxcr4 and those for Itgb1 were transferred into pT2K-CAG-DEST (see below) and PB-CA vector (a gift from Andras Nagy, Addgene plasmid #20960)[73] using Gateway LR Clonase Enzyme Mix (Invitrogen), respectively. The transposon vector, pT2K-CAG-DEST, was constructed by swapping the EGFP cDNA (EcoRI/BglII sites) of pT2K-CAGGS-EGFP (see above) with Reading Frame Cassette A (RfA, Invitrogen, Gateway Vector Conversion System, Cat# 11828-029) flanked with attR1 and attR2 sequences (Invitrogen, Gateway system). pCAG-Cxcr4-myc was constructed by inserting the mouse Cxcr4 cDNA from pMD-mCXCR4 (Sino Biological Inc.) into pCAG = MCS2 with Myc-tag (N-terminal), a gift from T. Kawauchi[74]. pCAG-Itgb1-myc was constructed by introducing Myc-tag into the C-terminal of mouse Itgb1 coding sequence of pCAG-Itgb1[75]. To generate pPB-CAG-Itgb1-R, we first inserted annealed synthetic DNA (5'-CTAGCaGGTACCaCTCGAGaGAATTCtGGATCCa, 5'-TCGATGGATC CAGAATTCTCTCGAGTGGTACCTG) into NheI and BamHI sites of pPB-CAG.EBNXN to generate pPB-CAG-NKXEB, then we ligated the SpeI-BamHI fragment of pCAG-Itgb1-myc to pPB-CAG-NKXEB to obtain pPB-CAG-Itgb1-myc. We introduced 7 synonymous mutations to this vector at the target sequence of Itgb1 KD#1 vector. Two PCR products were obtained from pPB-CAG-Itgb1-myc using two primer sets (FW1; 5'-AACGCGGCGGAGCCCGGGCCGC-3', RV1; 5'-CTAGATTACCGGATAT CCGTTGCTGACCAACAAGTTC, FW2; 5'-TATCCGGTAATCTAGACT CTCCAGAAGGTGGCTTTGA, RV2; 5'-GAGTTGTAGGCATCGATGATTAG-3'). These two PCR products have an overlapped sequence at the mutation site, and were inserted into SmaI and ClaI sites of pPB-CAG-Itgb1 by using Gibson method[68].

**Oligonucleotides inserted into pX330 (5'–3').** N-control (control vector); sense: caccgaaatgtgagatcagagtaat, antisense: aaacattactctgat ctcacatttc

Cxcr4 CR#1; sense: caccggttgacagtgtagatgata, antisense: aaactatc atctacactgtcaacc

Cxcr4 CR#2; sense: caccgtaccggtccaggctgatga, antisense: aaactc atcagcctggaccggtac

Cxcr7 CR#1; sense: caccgtcaaacaagtgcacatcca, antisense: aaactgg atgtgcacttgtttgac

Cxcr7 CR#2; sense: cacctaacagcagcgactgcattg, antisense: aaaccaa tgcagtcgctgctgtta

Olig2 CR; sense: caccgacctccgacgccaagtgagc, antisense: aaacgctc acttggccgtcggaggtc

Dcx CR; sense: caccgctgtggttccaccaaaata, antisense: aaactatttt ggtggaaccacagc

**Oligonucleotides for RR-puro vectors.** pPB-CAG-RR-Puro-Cxcr7 #1; sense: agaccatggatgtgcacttgtttgactatacgt,

antisense: atagtcaaacaagtgcacatccatggtctagct

pPB-CAG-RR-Puro-Cxcr7 #2; sense: atgtaacagcagcgactgcattgtggtg gacgt,

antisense: ccaccacaatgcagtcgctgctgttacatagct

pPB-CAG-RR-Puro-Dcx; sense: gttgctgtggttccaccaaaatatggaacta cgt,

antisense: agttccatattttggtggaaccacagcaacagct

**Oligonucleotides for miRNA vectors.** Itgb1 KD#1; sense: tgctgaatcca agtttccagatatgcgtttggccactgactgacgcatatctaaacttggatt

antisense: cctgaatccaagtttagatatgcgtcagtcagtggccaaaacgcatatctg gaaacttggattc

Itgb1 KD#2; sense: tgctgtaagccattagacctatcacagttttggccactgactga ctgtgatagctaatggctta

antisense: cctgtaagccattagctatcacagtcagtcagtggccaaaactgtgatagg tctaatggcttac

Cxcr4 KD#1; sense: tgctgcaaagtaccagtcagccatgggtttggccactgact gacccatggctctggtactttg,

antisense: cctgcaaagtaccagagccatgggtcagtcagtggccaaaacccatggc tgactggtactttgc

Cxcr4 KD#2; sense: tgctgtttccttggcctctgactgttgttttggccactgactg acaacagtcaggccaaggaaa,

antisense: cctgtttccttggcctgactgttgtcagtcagtggccaaaacaacagtcag aggccaaggaaac

Cxcr4 KD#3, sense: tgctgaacaccaccatccacaggctagttttggccactgac tgactagcctgtatggtggtgtt

antisense: cctgaacaccaccatacaggctagtcagtcagtggccaaaactagcctgt ggatggtggtgttc

**In utero electroporation**

In utero electroporation was performed as described previously with slight modifications from the original report[13]. Body temperature during the entire procedure was maintained by a heat pad placed below the animal. The plasmid was adjusted to appropriate concentrations (see below) with HEPES-buffered saline (HBS, #51558, Sigma-Aldrich). Diluted plasmid containing 0.01% Fast Green solution was then injected into the lateral ventricles of intrauterine embryos, and electronic pulses (35 V, 50 msec, 4 or 5 times) were applied using an electroporator (CUY-21 or NEPA21; Nepa Gene, Chiba, Japan) with a forceps-type electrode (CUY650P5). The embryos were allowed to live within the uterine horn until the desired time of observation. The compositions of plasmids for injection are as follows.

Figure 1a: pCAG-LNL-EGFP (0.5 μg/μl) + pCAG-LNL-LynEGFP (1 μg/μl) + Nestin-Cre (0.3 μg/μl)

Figure 1d: pCAG-kikGR (2 μg/μl)

Figure 1f: hGFAP-Cre (0.5 μg/μl) + pPB-CAG-LNL-turboRFP (0.5 μg/μl) + Tα1-EGFP (0.5 μg/μl)

Figure 2c: pCAG-hyPBase (0.5 μg/μl) + pPB-CAG-LNL-turboRFP (0.5 μg/μl)

Figure 3a and Supplementary Fig. 6a–c: rDIO system = pPB-CAG-rDIO-EGFP (0.5 μg/μl) + hGFAP-Cre (1 μg/μl) + pDcx-Dre (1 μg/μl) + pCAG-hyPBase (0.5 μg/μl)

Figure 3e and f: pCAG-LNL-EGFP (0.5 μg/μl) + pCAG-LNL-LynEGFP (1 μg/μl) + Nestin-Cre (0.3 μg/μl)

Figure 5d and e: PB-CAG-EGFP (0.5 μg/μl)

Fog. 6a, b: Cxcr4/7 CR or pX330-N-control (1 μg/μl, in the case of combinations of Cxcr4 CR and Cxcr7 CR, 0.5 μg/μl each), + pPB-CAG-EGFP (0.5 μg/μl) + pCAG-hyPBase (0.5 μg/μl)

Figure 6d: Cxcr4/7 CR (1 μg/μl) + pPB-CAG-EGFP (0.5 μg/μl) + pCAG-hyPBase (0.5 μg/μl)

Figure 7a: {Cxcr4 CR#1 and Cxcr7 CR#2 (0.5 μg/μl each)} or pX330-N-control (1 μg/μl), + pPB-CAG-EGFP (0.5 μg/μl) + pCAG-hyPBase (0.5 μg/μl)

Figure 7d: Itgb1 KD#1 or #2 (1 μg/μl) and/or pCAG-Myc-Cxcr4 (1 μg/μl) + pCAG-hyPBase (0.5 μg/μl)

Figure 8a–c: Cxcr4 CR#2 + Cxcr7 CR#1 (0.5 μg/μl each) or Cxcr4 CR#1 + Cxcr7 CR#2 (0.5 μg/μl each) or pX330-N-control (1 μg/μl), + pPB-CAG-LNL-EGFP (0.5 μg/μl) + hGFAP-Cre (1 μg/μl) + pCAG-hyPBase (0.5 μg/μl) + pCAG-turboRFP (0.2 μg/μl)

Figure 8d–f: {Itgb1 KD#1, KD#2, or pPB-CAG-miRneg (control, 1 μg/μl each)} + hGFAP-Cre (0.5 μg/μl) + pPB-CAG-LNL-turboRFP (0.5 μg/μl) + pCAG-hyPBase (0.5 μg/μl). In the rescue experiment, pPB-CAG-Itgb1-R (1 μg/μl) was added in the above plasmid mixture.

Supplementary Fig. 1d: pCAG-EGFP (1 μg/μl)

Supplementary Fig. 3a: pPB-CAG-EGFP (0.5 μg/μl) + pCAG-hyPBase (0.5 μg/μl) + pCAG-turboRFP (0.2 μg/μl)

Supplementary Fig. 3f: pCAG-T2TP (1 μg/μl) + pT2K-CAG-mCherry (1 μg/μl)

Supplementary Fig. 4a: hGFAP-Cre (1 μg/μl) + pPB-CAG-LNL-GFP (0.5 μg/μl)

Supplementary Fig. 5: pCAG-LNL-EGFP (0.5 μg/μl) + pCAG-LNL-LynEGFP (1 μg/μl) + Nestin-Cre (0.3 μg/μl)

Supplementary Fig. 8a: {Cxcr4 CR#1, CR#2, or pX330-N-control (1 μg/μl each)} + pT2K-CAG-EGFP (0.5 μg/μl) + pCAG-T2TP (1 μg/μl)

Supplementary Fig. 9b: T2K-Cxcr4 KD#3 (1 μg/μl) + pCAG-T2TP (1 μg/μl).

## Time-lapse imaging of brain slices

Time-lapse imaging was performed as described previously with slight modifications[11,12]. The brains were dissected and embedded in 3% low melting point agarose (SeePlaque, FMC) in Hanks' balanced salt solution (HBSS) containing 0.14 g/L CaCl$_2$ and 0.2 g/L MgSO$_4$−7H$_2$O. Next, the brains were sectioned coronally at 250 μm using a vibrating microtome (VT1000, Leica or HM650-V, Zeiss or Thermo Scientific). Brain slices obtained from around the level of the interventricular foramen were placed on a Millicell-CM (pore size, 0.4 μm; Millipore) placed in a glass-bottomed dish (Iwaki, Japan) containing Neurobasal medium (Life Technologies) with 10% fetal bovine serum (FCS), 2% B27 (Life Technologies), 50 units/mL penicillin, and 50 mg/mL streptomycin. The culture dish was then placed in an incubator chamber (5% CO$_2$ and 40% O$_2$ at 37 °C, Tokai Hit) fitted onto a confocal microscope (FV1000; Olympus). Approximately 8–15 optical Z sections for 6 to 24 positions were obtained automatically every 10 to 30 min, and about 10 focal planes (~ 50 μm thick) were merged to visualize the entire cell shape and trajectory. The movie files were constructed using 4D-Viewer (Ratoc System Engineering).

## Quantitative analysis of cell movement in brain slices

To trace the trajectories of individual cells, we used MTrackJ, a plug-in for ImageJ software (National Institutes of Health; local cursor snapping mode was activated, Figs. 1a, b, 2a, 6d, Supplementary Fig. 1a, b). The frames in which the cells had moved at a speed of more than 10 μm/h were subjected to analysis. The directional changes in migration (Supplementary Fig. 1a) were determined by subtracting the moving directions of two consequent frames. Positive values of differential degree mean rightward turns in the right hemisphere on slices, or leftward turns in the left hemisphere. The angles of the moving directions relative to the radial fiber (Supplementary Fig. 1b) were calculated by subtracting the moving direction of the cell and the direction of the radial fiber as visualized using GFP. The lateral

direction and medial direction of the brain slices were represented as plus and minus values, respectively. For the analysis of blood vessel-guided migration (Fig. 6d), the cells that could be traced on blood vessels continuously at least 5 h were selected and their maximum distances from the start points during the 24 hour-observation were evaluated using MTrackJ. We also performed computer assisted automatic trace using TrackMate plug-in of ImageJ[76] (Figs. 1g, 3e, 6c, Supplementary Fig. 3f). In Fig. 1g, GFP$^-$/RFP$^+$ cells were extracted by subtracting GFP signals from RFP signals, and then GFP$^+$ cells and GFP$^-$/RFP$^+$ cells were separately traced. As to Fig. 3e, we superimposed the images of GFP (astrocyte progenitors) and RFP (Flt1-DsRed, blood vessels) at each time frame, and then separated the GFP signals into those that were fused with RFP signals (vessel-associated) or not (vessel-independent), and traced them separately. As to Supplementary Fig. 3f, mCherry signals were traced. We counted the cells whose displacements were over 20 μm during the first 24 hour-observations.

## Determination of the final fate of erratically migrating cells in vitro

E15 or E16 mouse brains were electroporated with pCAGGS-kikGR (1 μg/μl) and slices were prepared on the next day. During the time-lapse observations, we used a 405-nm laser to irradiate erratically migrating cells or radial migrating cells to convert their fluorescence from green to red on the independent slices. Then, tissue fragments of slices that included red cells were treated with Trypsin and DNaseI and were dissociated into single cells. The cells were then seeded onto poly-L-lysine-coated cover slips printed with a numbered grid pattern (GC1300; Matsunami) and cultured in Neurobasal medium containing 10% FCS in 5% CO$_2$ at 37 °C. During the cultivation, the red fluorescence gradually changed back to green because of the degradation of converted kikGR protein and de novo synthesis from the introduced plasmid. We therefore recorded the positions of the red fluorescent cells on the grid on the first day of culture and re-identified the cells located at the same positions every 24 h and irradiated them with the 405-nm laser to maintain the red color. After 4 days in vitro, the cultured cells were fixed with 4% PFA, and were subjected to double immunocytochemistry with primary antibodies, TuJ1 (mouse, BioLegend, Covance, Cat# 8578, 1:200) and anti-GFAP (rabbit, Dako, Cat#Z0334, 1:400), followed by secondary antibodies, Cy5-conjugated anti-rabbit IgG (donkey, Jackson ImmunoResearch, Cat#711-175-152, 1:100) and DyLight405-conjugated anti-mouse IgG (donkey, Jackson ImmunoResearch, Cag#715-475-151, 1:100). During the culture period, irradiated cells sometimes divided and formed colonies, which was especially seen in the culture of the erratically migrating cells. We therefore counted the number of colonies, instead of cells.

## Cell fate analysis in vivo using Olig2-CreER mice

Olig2-CreER mice[24] and the Z/EG reporter line[25] were crossed, and obtained Olig2-CreER;Z/EG double heterozygote male mice were then crossed with female ICR mice. E15 mouse embryos (1:4 of them are theoretically Olig2-CreER;Z/EG) were electroporated with pPB-CAG-LNL-turboRFP (0.5 μg/μl) together with pCAG-hyPBase (0.5 μg/μl), and applied 200 μL of tamoxifen solution (20 mg/ml in corn oil) by gavage 2 days after the in utero electroporation. At P30, the manipulated mice were fixed and coronally sliced in 100 μm thick, and employed to immunohistochemistry of Aldh1l1, GSTπ, and GFP.

## Tild-CRISPR method

Tild-CRISPR method to analyze the lineage of Olig2 positive cells derived from the cortical VZ was done following a previous paper[77]. Tild donor double strand DNA was prepared as described below. Genome DNA of Olig2 was subcloned into pLSODN vector (BioDynamics), and replaced its stop codon with a 2A-Cre cassette from Cas9-2A-Cre, which was a gift from Su-Chun Zhang (Addgene plasmid #68468)[78]. We performed PCR using forward (5'-cagcggcttcacaggaggga) and reverse

(5'-tgcgtgagtgtgtgtgtgcg) primers, which encompass from 800 bp upstream to 800 bp downstream of the stop codon, and purified with silica columns (NulceoSpin, Macherey-Nagel) and then performed ethanol precipitation. We electroporated Olig2 CR (1 µg/µl), Tild donor (0.25 µg/µl), pCAG-hyPBase (0.5 µg/µl), pPB-CAG-LNL-EGFP (0.3 µg/µl) and pCAG-turboRFP (0.2 µg/µl) into E16 mouse brains in utero, and fixed at P30.

## In vivo two-photon imaging

In vivo two-photon imaging was performed as described previously[79,80]. We electroporated *Flt1-DsRed* mice with rDIO system by using in utero electroporation at E15 (Fig. 4a). Two days later, the pregnant mice were anesthetized by inhalation of isoflurane (initially 2% [partial pressure in air] and then reduced to 1%). Body temperature of the anesthetized dam mouse during the entire procedure was maintained by a heat pad placed below the animal. Depth of anesthesia was assessed by monitoring the tail-pinch reflex and respiration rate. The uterine horn was exposed and made small incision on it to take out the electroporated embryos with the umbilical cord connected to the dam. The embryos were placed in a small container and the body and head were fixed in the container with ultraviolet curing resin as shown in Fig. 4a. During this procedure, the temperature of the embryo was maintained around room temperature by placing the container of the embryo in a chamber filled with the ACSF solution of which temperature was maintained at room temperature. After curing the resin, skin was carefully removed and a cranial window was made above the S1 region where rDIO plasmids were introduced. The craniotomy sites were covered by a glass coverslip (4 × 4 mm, Matsunami Glass, Osaka, Japan), which was briefly attached to the skull using ultraviolet curing resin and fixed with dental cement. DsRed expressing blood vessels and GFP positive astrocyte progenitors were observed using a two-photon microscope (A1R MP, Nikon; Tokyo, Japan) with a water immersion objective lens (16x, 0.80 N.A., Nikon). Images were taken from $795 \times 795 \, \mu m^2$ square region at a depth of 0 to 387 µm under the brain surface (1 µm z-step, sequentially scanned from the surface, under the regulation of the z-axis piezo nanopositioner) at 1 min intervals for about 4 h consecutively. Blood flow of the embryonic brain was also monitored to examine viability of embryos during recording. Images were analyzed with ImageJ software (Molecular Devices, Downingtown, PA, USA) and MATLAB software version R2014a (MathWorks Inc., MA, USA).

## Analysis of Single-cell RNA-seq data

Single-cell RNA-seq data from E18 mouse whole brains were obtained from 10X Genomics (https://support.10xgenomics.com/single-cell-gene-expression/datasets/1.3.0/1M_neurons. Providing contact information is needed for downloading); 1,308,421 (1.3 million) individual cells from embryonic mouse brains were sequenced and profiled using Cell Ranger 1.3.0 protocol. Seurat package version 3.2.2[81] in R (version 4.0) was utilized for analysis of randomly selected 53,541 cells from the 1.3 million whole brain single-cell RNA-seq database. Data in each cell were normalized by NormalizeData function (method = " LogNormalize", scale.factor = 10000). The 1000 most variable genes were identified by FindVariableFeatures (selection.method = "vst") function. The expression levels of the genes were scaled by ScaleData function before performing principal component analysis (PCA). The functions in the Seurat package, RunPCA, FindNeighbors, FindClusters, and RunUMAP, were performed with default parameters for identifications of clusters and dimensionality reduction. Finally, potential marker genes for the clusters were computed by FindMarkers and FindAllMarkers functions. The differentially expressed genes (DEGs) were determined by the threshold of adjusted P-value <0.01 for the probability of differentially expression between astrocyte progenitors and migrating neurons. To perform GO analysis, we uploaded list of astrocyte progenitor-enriched DEGs (4723 genes) and

migrating neuron-enriched DEGs (2122 genes) to the DAVID Bioinformatics Resource (https://david.ncifcrf.gov/home.jsp) and used the GOTERM_BP_DIRECT annotation. To investigate the difference in cell adhesion molecules between astrocyte progenitors and migrating neurons, we selected DEGs annotated with "cell adhesion" (GO:0007155), uploaded the gene lists to the DAVID, further annotated protein domains using SMART option (false discovery rate <0.05), and classified them based on the protein structures.

## in situ HCR

To detect *Cxcr4*, *Ackr3(Cxcr7)* and *Olig2* mRNA, we used third-generation in situ hybridization chain reaction (HCR)[82] following the method described previously[83]. We electroporated with pPB-CAG-EGFP at E15, and perfused with 4% paraformaldehyde (PFA) in 0.1M phosphate buffer at E18. The brains were dissected out and sliced into 100 µm thick coronal sections. After washing with PBS, the slices were incubated in a hybridization solution (Molecular Instruments, Los Angeles, CA) at 37 °C. The probe set for mouse *Cxcr4* (NM_001356509.1), *Ackr3* (NM_001271607.1), and *Olig2* (NM_016967) were designed and purchased from Molecular Instruments. Brain slices were incubated with 4-nM probes overnight at 37 °C (Olig2 probe was mixed with Ackr3 probe for double staining). After washing with a wash solution (Molecular Instruments) followed by 5x SSC with 0.1% Tween20, the slices were incubated with fluorescence-labeled hairpins (Alexa Fluor 594 for Cxcr4 and Ackr3, and Alexa Fluor 647 for Olig2, Molecular Instruments) overnight. The slices were washed with 5xSSCT, and then employed to immunohistochemistry for Olig2 (for sections of Cxcr4 HCR) or DAPI staining (for Ackr3 HCR) as described in the method of immunohistochemistry.

## Inhibition of Cxcr4 expression by CRISPR vectors in primary culture of astrocytes

E16 mouse embryos were electroporated with CRISPR vectors, Cxcr4 CR#1 or CR#2 (1 µg/µl each), together with a transposon EGFP expression vector system, pT2K-CAG-EGFP (0.5 µg/µl) + pCAG-T2TP (1 µg/µl). On the next day, the transfected brains were taken out, dissociated with 0.1% Trypsin, and cultivated on poly-D-lysin coated cover slips with culture medium; Neurobasal medium (Life Technologies) with 10% FCS, 2% B27 (Life Technologies), 50 units/mL penicillin, and 50 mg/mL streptomycin. After 4 days, the cells were fixed with 4% PFA, and incubated with rabbit anti-Cxcr4 (Abcam, ab124824, clone UMB2, 1:300) and mouse anti-GFAP (Sigma, G3893, clone G-A-5, 1:300), and then Alexa Fluor 555-conjugated anti-rabbit IgG (donkey, Invitrogen, A31572, 1:1000) and Alexa Fluor 647-conjugated anti-mouse IgG (goat, Invitrogen, A21236, 1:1000).

## Evaluation of the gene editing efficiency of CRISPR vectors for *Cxcr7*

To evaluate the gene-editing efficiency of CRISPR vectors for *Cxcr7*, we used the pCAG-EGxxFP reporter plasmid, which express GFP in response to the Cas9 mediated double-strand breaks at the target sequence[67]. COS7 cells were co-transfected with pCAG-EGxxFP-Cxcr7, pCAG-mCherry and CRISPR vectors (Cxcr7 CR#1, CR#2, or control vector) using Polyethyleneimine, fixed with 4% PFA 2 days later, and GFP and mCherry fluorescent images were acquired using a fluorescent microscope (BZ-9000, Keyence). The GFP+ rate in mCherry+ cells were evaluated using ImageJ software.

## Indel rates of CRISPR vectors for *Cxcr4* and *Cxcr7*

E15 or E16 ICR mouse brains were electroporated with CRISPR vectors for *Cxcr4* or *Cxcr7* (Cxcr4 CR#1, #2, Cxcr7 CR#1, #2 or Dcx CR as a negative control, 1 µg/µl), together with PiggyBac transposon vectors (pCAG-hyPBase; 0.5 mg/ml, pPB-CAG-EGFP; 0.5 µg/µl), and puromycin resistant gene expression vectors (PB-CA-puro [0.5 µg/µl] for Cxcr4 CR, or gene-editing dependent puromycin resistance gene expression

vectors harboring the corresponding target sequences of Cxcr7 CR and Dcx CR vectors; pPB-CAG-RR-Puro-CxcR7 #1, #3, and pPB-CAG-RR-Puro-Dcx, [0.5 µg/µl]). One day later, the GFP positive area of the electroporated brains were dissected out manually under a fluorescent dissection microscope (MVX10, Olympus), dissociated with trypsin, and employed to neurosphere culture in DMEM/F12 supplemented with 2% B27, 20 ng/ml bFGF and 20 ng/ml EGF. Three to four days later, the cells were transferred to selection medium containing 1 µg/ml puromycin and cultured for 8 days. The genome DNA was extracted from these cells and amplified target region by PCR using specific primers (5'-TAGAACTAGTGGATCCctgatttcacacctaggacacagcca and 5'-GCTTGATATCGAATTccaagatgtagcagtgcgtgtcgtag for Cxcr4, 5'-TAG AACTAGTGGATCCctgatttcacacctaggacacagcca and 5'-GCTTGATATC GAATTccaagatgtagcagtgcgtgtcgtag for Cxcr7, the overhangs for Gibson assembly were represented in uppercase). The PCR product was cloned into pBluescript II vector using Gibson method[68] and 16 independent clones were randomly selected and sequenced.

## Distances from astrocyte progenitors to nearest blood vessel

The brains were sliced coronally (100 µm thick) using a vibrating microtome. The sections were then subjected to immunohistochemistry for Olig2 or Aldh1l1 and GFP as described in the "Immunostaining" section. The blood vessels were stained with biotin-conjugated isolectin B4 (Sigma, 1:300) and Alexa Fluor 555 (Molecular Probes)-conjugated streptavidin (1:500). About 50 optical Z sections with a 1-µm interval were obtained using a confocal microscope. The image of the vasculature was skeletonized and re-constructed with voxels using 3D Object Counter, an ImageJ plugin. The distributions of the GFP-positive Olig2 or Aldh1l1-positive cells were also re-constructed independently in the same way. These two data sets were then merged using 3D RoiManager, an ImageJ plugin, and the distances from the nuclear staining in the double-positive cells to the nearest blood vessel (center to surface) were calculated using this software. The double-positive cells in the outer limits of the 15-µm sections (longer than the average distance from DAPI stained non-selected cell nuclei to the nearest blood vessel) and the marginal 20 µm of the XY plane were excluded from the analysis because closer blood vessels could have been cut off from the analyzed 3D space. As a reference, the distance between DAPI stained non-selected cell nuclei and nearest blood vessel were measured using the same method. DAPI staining of Z stack images from 15 to 20 µm was used.

## Cultivation of glia progenitors on primary endothelial cell-tubes and its time-lapse observation

E15 ICR mouse brains were electroporated with CRISPR vectors (Cxcr4 CR#1, Cxcr7 CR#1, #2, and Dcx CR, 1 µg/µl) and PiggyBac vectors (pCAG-hyPBase and pPB-CAG-EGFP, 0.5 µg/µl each). The GFP+ regions of brains were sampled the next day, and the neurospheres were cultured as described above. After one to six passages (3 to 15 days) of the neurosphere culture, the cells were transferred to glia differentiation medium (neurosphere culture medium + 10% FCS). Within the next 48 h, they attached to the bottom of petri dish and became GFAP+, but still maintained their high ability to proliferate and migrate. We harvested them to cultivate on endothelial cell-tubes. Primary endothelial cells were prepared as described in a textbook[84]. Briefly, E18 to P2 ICR mouse brains were dissected out and minced into small tissue fragments mechanically using fine forceps and treated with 0.1% trypsin in PBS for 15 min at room temperature. Then we added equal volume of DMEM + 10% FCS, pipetted 3 to 10 times, placed one minute, and transferred cell suspension to another tube. We repeated this process 5 to 6 times until the tissue fragment became invisible. The collected suspensions containing small fragments of microvasculature were placed on petri dishes pretreated with a coating medium (Quick Coating Solution cAP-01, Angio-Proteomie). We removed unattached tissues or cells 3 h later, and started cultivation in a special medium for

endothelial cells (Medium 200 with LSGS, Thermo Fisher). After several passages, the endothelial cells were harvested and seeded on Matrigel (Corning) at the concentration of $1.5 \times 10^4$ cells/cm$^2$. By 24 h after the seeding, the endothelial cells formed a tube structure. The glia progenitors prepared as described above were inoculated on the endothelial tube at the concentration of $1.5 \times 10^4$ cells/cm$^2$. Time-lapse observation of the migration of glia progenitors was started the next day of the combination. The co-culture was kept in the medium for endothelial cells. To analyze the movement of glia progenitors on the primary endothelial cell-tubes (Fig. 6c), we superimposed the GFP signals and the contours of endothelial tubes extracted from transmission images, and separated the GFP signals those that were fused with images of endothelial tubes (shown in green) or not (shown in red), and traced them automatically using TrackMate plug-in of ImageJ. GFP+ cells attached to and separated from the endothelial tubes were identified by using the same method for slice culture of *Flt1-DsRed* mouse brains described above. The cells that could be continuously traced for 10 h during the 24 hour-observations were counted, and the rates of migrating off the endothelial tubes against migrating on them were evaluated in each objective field.

## Astrocyte distribution in the CP

To analyze the final positioning of astrocytes generated in the restricted period within the VZ, we electroporated with GFP expression PiggyBac transposon vector (pCAG-hyPBase, transposase expression vector; pPB-CAG-LNL-EGFP or pPB-CAG-LNL-turboRFP, CAG promoter driven Cre dependent EGFP or turboRFP expression cassette flanked by PiggyBac element for integration into the host genome; hGFAP-Cre or nestin-Cre, human GFAP promoter or nestin promoter driven Cre expression vector). The manipulated brains were fixed at P10 and prepared slices in 100 µm thick. The positions of GFP or RFP positive astrocytes (as Aldh1l1+) in the slices were automatically identified by using Image Calculator (a function of ImageJ, GFP or RFP channel x Aldh1l1 channel) and 3D maxima finder with size filter (larger than 6,000 µm$^3$, ImageJ plug-in), and their relative positions to the upper and lower border of the CP were computed. For knockdown experiment described in Supplementary Figure 9, E16 mouse embryos were electroporated with T2K transposon-based KD vector system (pCAG-T2TP, transposase expression vector; pT2K-CAG-EmGFP-miR-Cxcr4, EmGFP and miRNA expression cassette under the control of CAG promoter flanked by Tol2 element for integration into host genome), and fixed at P4. In this labeling method, neurons in the superficial most part of layer II/III were also labeled, making difficult to identify astrocytes within this region. We, therefore, excluded this region from the analysis. We divided the CP from the bottom of layer VI to the bottom of the accumulation of GFP+ neurons in layer II/III into four bins and the number of Olig2/GFP double positive cells in each bin were counted manually.

## Western blotting

To evaluate the knockdown efficiency of miRNA vectors for Cxcr4, COS7 cells were transfected with miRNA expression vector (Cxcr4 KD#1-3, or control vector) together with pCAG-Myc-Cxcr4 and pCAG-mCherry using Polyethyleneimine (Polyscience, Polyethyleneimine "Max")[85]. Two days later, the transfected cells were harvested and subjected to SDS-PAGE (10% gel) and Western blotting as described previously[86]. Rabbit polyclonal antibody against the Myc-tag were generated as described[86]. RFP expression was detected as the internal control (Rockland, 1:5000). The chemiluminescence was detected using LAS4000 mini (GE Healthcare), and the band intensities were used for quantitative analysis. The evaluation of knockdown efficiency of miRNA expression vector for Itgb1 was carried out on the same way. Neuro2a cells were transfected with miRNA vector for Itgb1 (Itgb1 KD#1, KD#2 or control vector) together with pCAG-Itgb1-myc. Expression of β-actin was examined as a loading control using anti-β-actin (mouse, Cell Signaling, 1:5000).

## Flow cytometry

Mice were sacrificed and the brains were rapidly dissected out. After mincing into 1–2 mm pieces, the cerebral hemispheres were incubated for 30 min at 37 ˚C in the presence of 10 unit/ml papain and 0.2 mg/ml DNase. Following filtration through a 70-μm pore size nylon mesh and removal of the papain solution, 30% Percoll was added and the cell suspension was centrifuged for 30 min at 500×g. The cell pellet was used for the Flow cytometry analyses. Cells were stained with the following antibodies: APC conjugated anti-ACSA-2 (Miltenyi Biotec, Cat#130-116-245, clone REA969, 1:100) and FITC conjugated anti-CD29 (BD Biosciences, Cat#102205, clone HMβ1-1, 1:100). Cells were analyzed with Cytoflex S Flow cytometer (Beckman Coulter) and data were analyzed with FlowJo software version 10.8.1 (BD Biosciences).

## Statistics and Reproducibility

As to in utero electroporation analyses, we introduced one experimental vector and one appropriate control vector in the same litter. We repeated the same injection at least twice, and confirmed the reproducibility. Statistical analyses were performed with Prism software (GraphPad, version 7) or R statistical package (version 4.0.0). Violin and box plots were drawn by ggplot2 R package. In the box plots, the box represents the first to third quartile range (interquartile range). The vertical lines (whiskers) represent the 1.5 x interquartile range below and above the first and third quantiles, respectively. The horizontal thick lines and dots in the boxes represent the median level and the average, respectively. Normally distributed data (determined by F test) were analyzed with unpaired Student's t test (for equal variance) or Welch's t-test (for unequal variance) when comparing between two data groups. Nonnormally distributed data were analyzed with the nonparametric Mann-Whitney test. As for comparing among three or more data groups, Dunnett post hoc tests (for comparison to a control group) or Tukey post hoc tests (for comparison among every group) were done. A probability of $P < 0.05$ was considered statistically significant. All significant statistical results are indicated within the figures with the following conventions: $*P < 0.05$, $**P < 0.01$, $***P < 0.001$, $****P < 0.0001$.

## Reporting summary

Further information on research design is available in the Nature Research Reporting Summary linked to this article.

# Data availability

The source data underlying Figs. 1c, h, 2g, h, 6b–d, 7a, d, 8c, f, and Supplementary Figs. 1a, b, 2d, 3g, 4d, 7b–g, 8b, d,9a, b are provided as a Source Data File. Previously published single-cell RNA-seq dataset[36] used in our clustering and DEG analyses (Fig. 5 and Supplementary Fig. 7) is available from 10X Genomics (https://support.10xgenomics.com/single-cell-gene-expression/datasets/1.3.0/1M_neurons). For all other inquiries, please contact the corresponding authors. Source data are provided with this paper.

# Code availability

All custom codes used in this study are available from the corresponding authors upon request.

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

## Acknowledgements

We are grateful to C.G.Lobe for Z/EG mice, K. Kawakami and Y. Takahashi for pCAGGS-T2TP and pT2K-CAGGS-mCherry, A. Miyawaki for kikGR cDNA, J. Miyazaki for CAG promoter, U. Lendahl for nestin promoter, C. Cepko for pCALNL-DsRed, S. Miyagawa for codon usage-optimized Cre, Y. Mizutani and K. Yoshizaki for manipulation of *Flt1-DsRed* mouse embryos, Y. Gotoh and K. Nakashima for their help to breed the Aldh1l1-GFP mice, N. Hane, T. Nagano, I. Iwamoto and H. Arai for technical supports, and M. Yoshitomo and T. Kobayashi for their assistance in preparing two-photon imaging. We also thank N. Hiroi for technical supports in single cell RNA-seq data analysis. This work was supported by JSPS KAKENHI [Grant Number JP21K07309 to H.Tabata, JP20H05688, JP16H06482, and JP22K19365 to K.Nakajima, JP16H06280 (Platforms for Advanced Technologies and Research Resources "Advanced Bioimaging Support") to M.A. and J.N., JP21K06413, and JP18K06508 to T.H.], Takeda Science Foundation, Keio Gijuku Academic Development Funds, and Keio Gijuku Fukuzawa Memorial Fund to K.Nakajima.

## Author contributions

H.T., K. Nagata and K.N. designed the experiments. H.T. performed most of the experiments. M.S. contributed to live imaging and cell tracking shown in Fig. 1. M.A. and J.N. designed and conducted the in vivo two-photon imaging. H.S. analyzed single cell RNA-seq data. M.M. performed flow cytometry. H.T. and K.I. provided *Olig2-CreER* mice. M.E. provided *Flt1-DsRed* mice. Y.H. performed in situ HCR. K.H. and Y.I. conducted animal care of mutant mice in Keio University and ADDC, respectively. T.H. constructed plasmid vectors. M.N. wrote R scripts and conducted statistical analysis. H.I. performed in vivo electroporation in the postnatal stages. H.T. and K.N. wrote the manuscript.

## Competing interests

The authors declare no competing interests.
