## [Peer Review File · Nature Communications]

Erratic and Blood Vessel-Guided Migration of Astrocyte Progenitors in the Cerebral CortexREVIEWER COMMENTS

Reviewer #1 (Remarks to the Author):

In this study, Tabata and colleagues report novel migration modes, erratic migration and blood vessel-guided migration, of astrocyte progenitors in the developing cerebral cortex. The authors also show that protoplasmic astrocytes are generated during prenatal period from these highly motile progenitors, while fibrous astrocytes are generated during postnatal period from progenitors that do not move extensively. They also found that Cxcr4/7 and integrin β 1 regulate the blood vessel-guided migration, and that the interaction between astrocyte progenitors and blood vessels supports angiogenesis by bridging neighboring blood vessels.

These findings have profound implications for the field of neuroscience, since the mechanism of astrocyte positioning during brain development has been largely unknown. The authors have used multiple cutting-edge techniques to provide convincing results. I only have a few comments as listed below:

1. The authors claim that migrating astrocyte progenitors enhance branch formation of blood vessels. If so, do the authors observe a reduction in vessel density and branching associated with impaired astrocyte progenitor migration due to Cxcr4/7 knockout or integrin β 1 knockdown? Is it not necessary to consider such abnormal formation of blood vessels in quantifying the distance between blood vessels and migrating astrocyte progenitors?
2. Are there any changes or differences in the expression levels of Cxcr4/7 and/or integrin β 1 between astrocyte progenitors that undergo erratic migration and blood vessel-guided migration? At least, it would be useful to discuss how astrocyte progenitors switch between the two modes of migration.

Minor comments:

1. The text and legend in Fig. 1d state that electroporation with KikGR was performed at E15 to determine the fate of irregularly migrating cells. However, the Methods section states that electroporation was performed at E16, and the illustration in Fig. 1d says E15 or E16. Please check the correct date(s) and make sure they are consistent.
2. Please provide some details of the immunohistochemistry procedure in the Methods section. Currently, only antibody information has been provided.
3. In the Methods section, the URL for the E18 mouse whole brain single-cell RNA-seq dataset is simply the URL to the 10x genomics company website, not a specific dataset. Please provide the specific URL for the dataset used.

Reviewer #2 (Remarks to the Author):

In this outstanding study, Tabata and colleagues discover a new mode of cell migration in the developing cerebral cortex, largely found only at late embryonic and early postnatal stages, where cells migrate in a meandering manner. The authors identify that these cells are astrocyte progenitor cells, and that their apparently random migration actually has an important component of chemotaxis and interaction with blood vessels. This is demonstrated beautifully by live imaging in brain slices as well as in vivo in living embryos. A fascinating detail is that the authors further show that migrating astrocyte progenitors reciprocally contribute to guide the formation of the blood capillary network. Then the authors mine public datasets of single-cell transcriptomics of the developing mouse cortex and find the transcriptomic signatures of these astrocyte progenitors and blood vessels. This leads to the identification of CXCR4 and CXCR7 as the chemotaxis receptors in astrocytes progenitor cells, CXCL12 as the likely chemoattractant secreted by blood vessels, and also the beta subunit of beta integrin (Itgb) as co-factor of CXCR signaling. By performing several in vitro and in vivo experiments

manipulating this signaling axis, the authors show its involvement in the interaction between migrating astroglial cells and blood vessels. The study is extremely original, rigorously well performed, technologically top, and provides an entirely original and novel framework to understand the development of glial cells in the cerebral cortex.

Based on these strengths, and the soundness of their experimental executions, I highly recommend publication of this study in Nature Communications. Nevertheless, there are a small number of points that I think should be first addressed for this study to be fully ready:

In line 127, page 6, the authors conclude: "These observations confirmed that erratic migration was the unique migration mode of astrocyte progenitors". However, their observations confirm that erratic migration is used by astrocyte progenitors and not neurons, but not that it is the unique mode for astro prog.

Images in Fig 2e are of insufficient quality to demonstrate expression of markers in GFP+ cells.

In Supp Fig 2c, why is the cortex not completely filled with GFAP+ and S100b positive cells, as it should be given that these are markers of such cells?

The quality of images in Fig 4d,e is poor and largely insufficient to visualize what the authors pretend to convey.

Why the phenotype of astrocyte progenitor distancing from blood vessels stronger with Cxcr7 KD than with Cxcr4 KD in vitro, when in vivo it is the opposite?

In Fig 7c, Itgb KD did not rescue back to control levels the deleterious effects of Cxcr4 OE, but it increased the distance significantly compared to controls, a situation very similar to Itgb KD alone. With these results, the CXCR4-ITGB axis is not demonstrated, but only that Itgb can override the attractive influence of Cxcr4.

The last section of the manuscript is entitled: "The final position of astrocytes is affected by functional blocking of Cxcr4/7-Integrin β 1 signaling axis". This axis is not demonstrated here, as also not in the previous section (see previous comment), so this conclusion is not supported and needs revision. The authors' experiments do not show phenotype rescue before, and they did not perform phenotype rescue here at all.

Reviewer #3 (Remarks to the Author):

This manuscript by Tabata and colleagues describes the results of a comprehensive investigation on the details and mechanisms underlying migration of astrocyte progenitors in the developing cerebral cortex of a mouse. The authors show that astrocyte progenitors move either irregularly or adopt blood vessel-guided migration within the developing cortex. On the mechanism of guided migration the authors show that migration of astrocyte progenitors is dependent on Cxcr4/7-Itgb1 which is also essential for proper astrocyte positioning within the cortical parenchyma. The study appears to be expertly performed, the manuscript is very well written, and the data are beautifully illustrated. I only have one minor comment to make regarding the title of this paper. I would suggest to replace the word "dispersion" with "migration" and perhaps re-consider the use of the term "erratic". It must be a purpose in this behavioural migration pattern, perhaps "exploratory" might be a better term?

First of all, we thank all reviewers for their helpful and constructive comments. Here is our point-by-point response for the comments.

Reviewer #1 (Remarks to the Author):

In this study, Tabata and colleagues report novel migration modes, erratic migration and blood vessel-guided migration, of astrocyte progenitors in the developing cerebral cortex. The authors also show that protoplasmic astrocytes are generated during prenatal period from these highly motile progenitors, while fibrous astrocytes are generated during postnatal period from progenitors that do not move extensively. They also found that *Cxcr4/7* and integrin $\beta 1$ regulate the blood vessel-guided migration, and that the interaction between astrocyte progenitors and blood vessels supports angiogenesis by bridging neighboring blood vessels.

These findings have profound implications for the field of neuroscience, since the mechanism of astrocyte positioning during brain development has been largely unknown. The authors have used multiple cutting-edge techniques to provide convincing results. I only have a few comments as listed below:

1. The authors claim that migrating astrocyte progenitors enhance branch formation of blood vessels. If so, do the authors observe a reduction in vessel density and branching associated with impaired astrocyte progenitor migration due to *Cxcr4/7* knockout or integrin $\beta 1$ knockdown? Is it not necessary to consider such abnormal formation of blood vessels in quantifying the distance between blood vessels and migrating astrocyte progenitors?

This is a very interesting and critical point. Actually, we carefully observed the effects on the density of blood vessels in the *Cxcr4/7* CRISPR vector- or integrin beta 1 knockdown vector-electroporated brains, but we did not detect significant changes. This may be due to the fact that not all of the astrocyte progenitors were transfected by in utero electroporation, and untransfected cells may have compensated for the angiogenic effect. We have added this explanation in the manuscript (lines 324-329).

2. Are there any changes or differences in the expression levels of *Cxcr4/7* and/or integrin $\beta 1$ between astrocyte progenitors that undergo erratic migration and blood vessel-guided migration? At least, it would be useful to discuss how astrocyte progenitors switch between the two modes of migration.

This is also a very interesting and important point. We are now extensively analyzing this issue, but do not have clear answer yet. We have added discussion on this issue in lines 431-439.

Minor comments:

1. The text and legend in Fig. 1d state that electroporation with KikGR was performed at E15 to determine the fate of irregularly migrating cells. However, the Methods section states that electroporation was performed at E16, and the illustration in Fig. 1d says E15 or E16. Please check the correct date(s) and make sure they are consistent.

We are sorry for this mistake. We collected the data from the slices prepared from the brains electroporated at E15 and E16. We have corrected this point in the Result, legend of Fig. 1d, and the Method section.

2. Please provide some details of the immunohistochemistry procedure in the Methods section. Currently, only antibody information has been provided.

We thank this helpful comment. We have added the detailed procedure of immunohistochemistry in the Methods section (line 725 - 748).

3. In the Methods section, the URL for the E18 mouse whole brain single-cell RNA-seq dataset is simply the URL to the 10x genomics company website, not a specific dataset. Please provide the specific URL for the dataset used.

We thank this valuable comment. The direct URL for the dataset of 1.3 million E18 mouse whole brain single-cell RNAseq is as follows.
https://support.10xgenomics.com/single-cell-gene-expression/datasets/1.3.0/1M_neurons

We have replaced the URL in the Method section (line 972), however, contact information, including names, email address, institutions, and country, is necessary to access the page and download the data.

Reviewer #2 (Remarks to the Author):

In this outstanding study, Tabata and colleagues discover a new mode of cell migration in the developing cerebral cortex, largely found only at late embryonic and early postnatal stages, where cells migrate in a meandering manner. The authors identify that these cells are astrocyte progenitor cells, and that their apparently random migration actually has an important component of chemotaxis and interaction with blood vessels. This is demonstrated beautifully by live imaging in brain slices as well as in vivo in living embryos. A fascinating detail is that the authors further show that migrating astrocyte progenitors reciprocally contribute to guide the formation of the blood capillary network. Then the authors mine public datasets of single-cell transcriptomics of the developing mouse cortex and find the transcriptomic signatures of these astrocyte progenitors and blood vessels. This leads to the identification of CXCR4 and CXCR7 as the chemotaxis receptors in astrocytes progenitor cells, CXCL12 as the likely chemoattractant secreted by blood vessels, and also the beta subunit of beta integrin (Itgb) as co-factor of CXCR signaling. By performing

several in vitro and in vivo experiments manipulating this signaling axis, the authors show its involvement in the interaction between migrating astroglial cells and blood vessels. The study is extremely original, rigorously well performed, technologically top, and provides an entirely original and novel framework to understand the development of glial cells in the cerebral cortex.

Based on these strengths, and the soundness of their experimental executions, I highly recommend publication of this study in Nature Communications. Nevertheless, there are a small number of points that I think should be first addressed for this study to be fully ready:

1) In line 127, page 6, the authors conclude: “These observations confirmed that erratic migration was the unique migration mode of astrocyte progenitors”. However, their observations confirm that erratic migration is used by astrocyte progenitors and not neurons, but not that it is the unique mode for astro prog.

The reviewer is correct. We have changed all the expressions “unique” to “characteristic” including the above sentence.

2) Images in Fig 2e are of insufficient quality to demonstrate expression of markers in GFP+ cells.

We understand the reviewer’s concern about this. However, there is a technical difficulty about this. In this experiment, we performed quadruple staining for GFP, RFP, Aldh111, and GSTpi. Aldh111 was stained with Alexa Fluor 405, and the high background was unfortunately inevitable upon acquisition of the images using a UV laser. We carefully adjusted the images and have replaced them in revised Fig 2e, but we have to admit the background is still rather high. We have added a statement indicating they are quadruple staining in the Figure legend and the Method section (line 743).

3) In Supp Fig 2c, why is the cortex not completely filled with GFAP+ and S100b positive cells, as it should be given that these are markers of such cells?

We actually could observe many S100b-positive astrocytes other than GFP-positive cells in the cortex. We have added arrowheads pointing to the GFP-positive and -negative cells among S100b-positive cells in Supplementary Fig 2c. On the other hand, GFAP-expressing astrocytes are quite few in the cortical gray matter, and we selected a picture including some GFP-positive and GFAP-positive cells for Supplementary Fig 2c, and in this picture, GFP-negative/GFAP-positive cells were not seen.

4) The quality of images in Fig 4d,e is poor and largely insufficient to visualize what the authors pretend to convey.

We have carefully reconstructed the images and replaced them. Now we believe that

the pictures have been somewhat improved. We also provided pictures showing DsRed channel alone to clearly show the relative position of GFP-positive cells to blood vessels. These revisions clearly show that some astrocyte progenitors migrate in vessel free space in the brains of living mouse embryos.

5) Why the phenotype of astrocyte progenitor distancing from blood vessels stronger with Cxcr7 KD than with Cxcr4 KD in vitro, when in vivo it is the opposite?

We think this is due to the difference in the sensitivity of different experiments (Fig 6b and c). Cxcr7 CR#1 was sufficient to disturb the vessel-guided migration in the in vitro culture system shown in Fig 6c. However, there is no significant difference between Cxcr4 CR#2 and Cxcr7 CR#1 in Fig 6c ($P > 0.999$, Dunn's multiple comparisons test). Therefore, we cannot say Cxcr7 CR is more effective than Cxcr4 CR in vitro. We have added this information in the legend.

6) In Fig 7c, Itgb KD did not rescue back to control levels the deleterious effects of Cxcr4 OE, but it increased the distance significantly compared to controls, a situation very similar to Itgb KD alone. With these results, the CXCR4-ITGB axis is not demonstrated, but only that Itgb can override the attractive influence of Cxcr4.

The “over rescue” effect by Itgb1 KD for Cxcr4 OE phenotype can be explained that the Itgb1-KD vector inhibits the function of not only overexpressed Cxcr4, but also intrinsic Cxcr4 by reducing the cellular content of Integrin $\beta 1$, which is supposed to act as the final effector for the vessel association. We have added this notion in the main text.

7) The last section of the manuscript is entitled: “The final position of astrocytes is affected by functional blocking of Cxcr4/7-Integrin $\beta 1$ signaling axis”. This axis is not demonstrated here, as also not in the previous section (see previous comment), so this conclusion is not supported and needs revision. The authors' experiments do not show phenotype rescue before, and they did not perform phenotype rescue here at all.

The reviewer is absolutely correct. To answer this comment, we performed several lines of experiments and found that the amount of Integrin $\beta 1$ on the plasma membrane of astrocyte progenitors that had been transfected with CRISPR vectors for Cxcr4 and Cxcr7 was significantly reduced compared to control. This observation suggests that Integrin $\beta 1$ is a downstream target of Cxcr4/7 signaling in astrocyte progenitors, although we could not conclude whether this effect is direct or indirect. We have added these data in Fig 7a. In addition, we have added a rescue experiment of Itgb1 KD by introducing KD-resistant Integrin $\beta 1$ (Fig 8e,f).

Reviewer #3 (Remarks to the Author):

This manuscript by Tabata and colleagues describes the results of a comprehensive

investigation on the details and mechanisms underlying migration of astrocyte progenitors in the developing cerebral cortex of a mouse. The authors show that astrocyte progenitors move either irregularly or adopt blood vessel-guided migration within the developing cortex. On the mechanism of guided migration the authors show that migration of astrocyte progenitors is dependent on *Cxcr4/7-Itgb1* which is also essential for proper astrocyte positioning within the cortical parenchyma. The study appears to be expertly performed, the manuscript is very well written, and the data are beautifully illustrated. I only have one minor comment to make regarding the title of this paper. I would suggest to replace the word "dispersion" with "migration" and perhaps re-consider the use of the term "erratic". It must be a purpose in this behavioural migration pattern, perhaps "exploratory" might be a better term?

We thank the reviewer to raise this issue regarding the title of this article. We perfectly agree to replace the word "dispersion" with "migration". As to "erratic", we agree that it is better replaced with some more intelligible and widely used word. However, although it is likely, we do not know whether these cells indeed "explore" something during this movement, because we didn't address the purpose of this behavior in this paper. We therefore would like to simply describe the movement without any interpretation. "Irregular movement/migration" or "random movement/migration" may be better to describe this behavior, but we are afraid that these names are too general as a name to specifically indicate the characteristic movement of the astrocyte progenitors, and the latter is not appropriate because the movement is not exactly random. We thus would like to propose to change the title as follows.

"Erratic and blood vessel-guided migration of astrocyte progenitors in the cerebral cortex."

REVIEWER COMMENTS

Reviewer #1 (Remarks to the Author):

The authors have nicely addressed all my comments. I have no additional comments.

Reviewer #2 (Remarks to the Author):

The authors have responded to all my comments and concerns satisfactorily, so in my opinion this manuscript is now acceptable for publication.

Point-by-point response to the reviewers' comments

First of all, we wish to express our appreciation to the reviewers for their helpful comments.

Reviewer #1 (Remarks to the Author):

The authors have nicely addressed all my comments. I have no additional comments.

Thank you for the positive comment.

Reviewer #2 (Remarks to the Author):

The authors have responded to all my comments and concerns satisfactorily, so in my opinion this manuscript is now acceptable for publication.

Thank you for the positive comment.